# OOD-Barrier: Build a Middle-Barrier for Open-Set Single-Image Test Time Adaptation via Vision Language Models

**Boyang Peng[1,2]**    **Sanqing Qu[1]***    **Tianpei Zou[1]**    **Fan Lu[1]**    **Ya Wu[3]**
**Kai Chen[1]**    **Siheng Chen[4]**    **Yong Wu[1]**    **Guang Chen[1,2]***

[1] Tongji University    [2] Shanghai Innovation Institute
[3] CNNC Equipment Technology Research (Shanghai) Co., Ltd.    [4] Shanghai Jiao Tong University

## Abstract

In real-world environments, a well-designed model must be capable of handling dynamically evolving distributions, where both in-distribution (ID) and out-of-distribution (OOD) samples appear unpredictably and individually, making real-time adaptation particularly challenging. While open-set test-time adaptation has demonstrated effectiveness in adjusting to distribution shifts, existing methods often rely on batch processing and struggle to manage single-sample data stream in open-set environments. To address this limitation, we propose Open-IRT, a novel open-set Intermediate-Representation-based Test-time adaptation framework tailored for single-image test-time adaptation with vision-language models. Open-IRT comprises two key modules designed for dynamic, single-sample adaptation in open-set scenarios. The first is Polarity-aware Prompt-based OOD Filter module, which fully constructs the ID-OOD distribution, considering both the absolute semantic alignment and relative semantic polarity. The second module, Intermediate Domain-based Test-time Adaptation module, constructs an intermediate domain and indirectly decomposes the ID-OOD distributional discrepancy to refine the separation boundary during the test-time. Extensive experiments on a range of domain adaptation benchmarks demonstrate the superiority of Open-IRT. Compared to previous state-of-the-art methods, it achieves significant improvements on representative benchmarks, such as CIFAR-100C and SVHN — with gains of +8.45% in accuracy, -10.80% in FPR95, and +11.04% in AUROC.

## 1 Introduction

In real-world environments, the model's ability to perform real-time adaptation is particularly crucial for handling the emergence of an unknown category or distributional shifts. This capability is essential in safety-critical applications such as autonomous driving, where failure to adapt can have serious consequences. Despite these demands, most approaches in computer vision [1–3] assume a closed-set paradigm, where training and testing data come from the same distribution. In contrast, real-world scenarios frequently encounter open-set environments [4–6], where models must handle unknown distributions and unseen categories, as shown in Fig. 1. This fundamental deviation from the closed-set assumption introduces fundamental challenges in maintaining model performance.

This transition from closed-world to open-set learning necessitates requiring innovative strategies to effectively filter, classify, and adapt to both distributional and semantic shifts without explicit supervision [7]. However, traditional approaches based on predefined categories are not equipped

---

*Corresponding Author

39th Conference on Neural Information Processing Systems (NeurIPS 2025).

to handle the emergence of unknown categories in open-set environments, particularly in real-time inference and single-sample adaptation constraints. As a result, accurate and efficient OOD detection has emerged as a critical research focus [7–9] in dynamic, open-set scenarios.

With the advent of vision-language models (VLMs), recent studies [10, 11] have leveraged their strong generalization for OOD detection tasks. Models such as CLIP [10] facilitate adaptation to unseen categories in open-set situation by learning rich cross-modal representations. Furthermore, recent research [12] have demonstrated that VLMs can perform zero-shot reasoning based on image-text associations and further adapt to single image inference, which enhances both their zero-shot generalization and OOD detection performance.

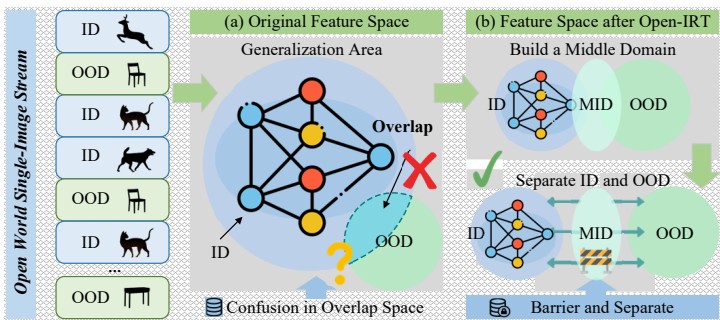

Figure 1: The objective of Open-IRT. In open-set environments, models encounter individual sample streams, which may include unknown categories. Existing test-time adaptation methods mainly focus on closed-set settings or batch image processing, neglecting single-sample data streams. (a) In such cases, the boundary between ID and OOD data can become unclear. (b) Our goal is to improve domain separation via a middle domain (MID) strategy, as shown in Fig. 4, and use test-time adaptation with VLMs to reduce ambiguity.

Test-time adaptation [13–16] offers a promising approach for adapting models to incoming data streams during inference, without relying on labeled data. Recent developments have extended this paradigm to single-sample test-time adaptation [17, 18], which is particularly valuable in domains such as security surveillance and industrial inspection, where real-time adaptation to dynamic environments is critical. However, most existing single-image test-time adaptation methods mainly focus on closed-set assumptions, whereas open-set batch processing strategies struggle to effectively adapt to changes in individual samples, particularly in real-time scenarios where only a single image is available for adaptation, as illustrated in Fig. 1.

To overcome this limitation, we focus on Open-Set Single-Image Test-time adaptation setting, where the model dynamically adapts to each input sample during inference without relying on batch data. Specifically, we propose a novel *Open-set Intermediate-Representation-based Test-time adaptation* (**Open-IRT**) framework for single-sample test-time adaptation in OOD detection. The overall motivation is illustrated in Fig. 1. As illustrated in Fig. 1b, Open-IRT establishes a structured separation between ID and OOD samples in feature space. The framework consists of two key modules. First, we introduce a *Polarity-aware Prompt-based OOD Filter* (**PPF**) module in Fig. 2a. Here, the term "polarity" refers to our utilization of the disparity between positive and negative prompts. PPF leverages the rich cross-modal information in vision language models to fully construct the ID-OOD distribution from both positive and negative prompts. It is guided by Semantic Contrast Hypothesis 1, which considers the absolute semantic alignment and relative semantic polarity. Next, we introduce *Intermediate Domain-based Test-time adaptation* (**IDT**) module in Fig. 2b based on Intermediate-Domain Hypothesis 2, which indirectly decomposes the distributional discrepancy between ID and OOD representations by modeling an intermediate domain that bridges the gap between these two. As shown in Fig.4, the IDT explicitly models this middle domain, and the learning objective enforces divergence of both ID and OOD samples from the intermediate domain. Therefore, this strategy effectively enlarging the distance between ID and OOD distributions indirectly. In addition, IDT uses a dynamic threshold strategy to generate bidirectional pseudo-labels, encouraging the model to reinforce positive feature representations and suppress intra-class noise.

**The main contributions are summarized as follows:** (i) We propose the PPF, an effective OOD filtering mechanism based on Semantic Contrast Hypothesis 1. The PPF captures both absolute semantic alignment and relative semantic polarity between an input and its paired prompts, enabling effective ID-OOD separation. (ii) We introduce the Intermediate-Domain Hypothesis 2, which leads to the development of IDT module. This module construct a intermediate domain strategy to establish a real-time two-way repulsion constraint between ID and OOD feature distributions, enhancing ID-OOD separation indirectly. (iii) Open-IRT consistently outperforms prior state-of-the-art methods

on standard benchmarks, including ImageNet-C [19], Tiny-ImageNet [20], VisDA [21], CIFAR-10C/100C [19], and digit datasets [22–24]. For instance, on CIFAR-100C and SVHN, it achieves +8.45% accuracy, -10.80% FPR95, and +11.04% AUROC, supporting our motivation and hypothesis.

## 2 Related Works

### 2.1 Out-of-Distribution Detection

Out-of-distribution (OOD) detection aims to determine whether a given sample originates from the training distribution or from an unseen distribution. The OOD detection approaches can be broadly categorized into classification-based methods [7, 25, 26] and density-based methods [8, 27]. Classification-based OOD methods relied on a maximum softmax probability to distinguish between ID and OOD samples. These include post hoc techniques such as ODIN [28], which utilizes temperature scaling and input perturbation. In addition, there are classic approaches such as JointEnergy scores [26], Mahalanobis distance [29], and activation space-based techniques [30, 31] that enhance the separability of ID and OOD samples without altering the training process. On the other hand, density-based methods use probabilistic models, such as class-conditional Gaussian distributions [29] and flow-based models [32, 9], to identify OOD samples based on their likelihood. To address high likelihoods challenges of OOD, techniques such as likelihood ratio [33], likelihood regret [34], and SEM scores [35] have been proposed. However, traditional methods are predominantly designed for the training phase, limiting their ability to adapt to distribution and semantic shifts in real-time.

### 2.2 OOD Detection with Vision Language Models

Vision-language models have gained significant attention in recent years for their ability to integrate visual and textual information. Renowned vision-language models, such as CLIP [10] and MaPLe [11], achieve impressive results by training on large-scale image-text pairs. To adapt vision-language models for downstream tasks (e.g., OOD detection), additional lightweight modules have been introduced, including prompt learners [36, 37], vision adapters [38, 39], and LoRA [40, 41].

Recent research in OOD detection has begun to utilize vision-language models as auxiliary tools, starting with CLIP [42], which aims to distinguish samples that do not belong to any ID class text provided by the user [43]. These approaches often employ techniques such as OOD label retrieval [44], generation [45], or alignment [46]. Training-free methods such as MCM [47] detect OOD using only ID labels, while auxiliary training-based methods, such as CLIPN [48] leverage additional pre-training to enhance OOD detection. From the perspective of specific training methodologies, some approaches implement a specialized handling of prompt words [49, 50], while NegPrompt [51] further explores the use of negative prompt techniques. Moreover, ROSITA [52] consider this task in test-time, yet it remains constrained by its reliance on direct feature alignment. Unlike previous methods, Open-IRT is fundamentally guided by an intermediate domain located near the boundary of the feature space, which indirectly decomposes the distributional discrepancy between ID and OOD representations. It introduces a bidirectional repulsion constraint, combining semantic alignment with relative semantic polarity, to increase the distributional separation.

### 2.3 Test-Time Adaptation

Test-time adaptation, originating from domain adaptation [53–55], adapts pre-trained models to test data with distribution shifts, without requiring training data. Test-time adaptation is essential for real-world applications, such as autonomous driving in diverse weather conditions. Several methods have been proposed, such as adjusting partial model weights [13, 56] or normalization statistics [14, 15]. Specifically, TENT [15] adapts batch normalization layers by entropy minimization, TTT [56] updates classification layers during testing, and T3A [13] introduces an optimization-free classifier adjustment. As a technology for changing environments, test-time adaptation can be integrated with OOD detection (e.g., RTL [57], UniEnt [58]). However, they typically address the task using batch processing methods, which limits their ability to handle dynamic scenarios effectively.

To reduce reliance on multiple test samples, some approaches prioritize single-sample adaptation. MEMO [59] enforces consistency through augmentations of the same test sample, while TPT [12] fine-tunes prompts for vision-language models during testing. DiffTPT [60] enhances this by using pre-trained diffusion models to augment test data. TDA [61] addresses computational efficiency

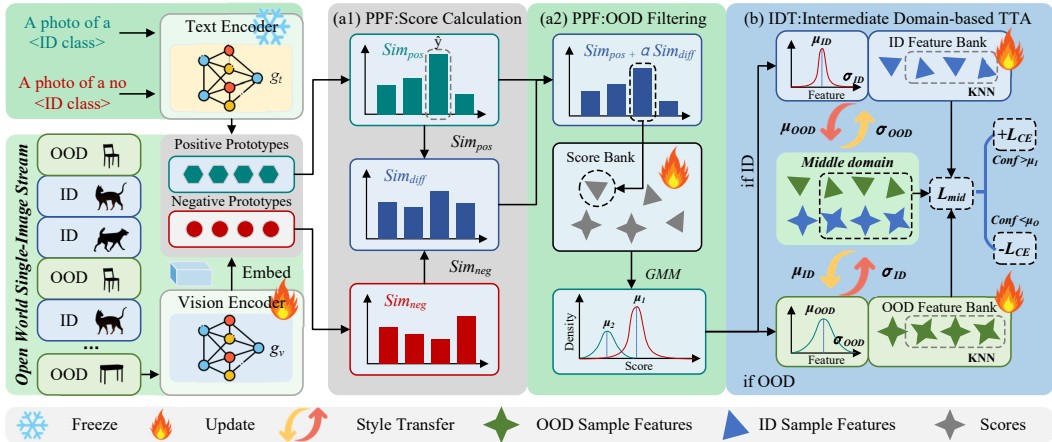

Figure 2: The architecture of Open-IRT comprises two key modules: (a) Polarity-aware Prompt-based OOD Filter and (b) Intermediate Domain-based Test-time Adaptation. (a) filters ID and OOD samples using positive and negative prompts, computing scores in Eq. 3 and utilizing a GMM strategy for OOD filtering. (b) generates an intermediate domain in Fig. 4 and Eq. 7 by leveraging mean and variance from feature banks, enhancing the model's adaptability with contrastive learning loss $\mathcal{L}_{mid}$ in Eq. 10 and pesudo-label loss $\mathcal{L}_{psd}$ in Eq. 13 during the test phrase.

with a cache-based model. While these methods assume closed-set settings, Open-IRT enables single-image Test-time adaptation in open-set environments. Unlike continual test-time adaptation, it further considers semantic shift in addition to distribution shift and refines the ID-OOD boundary in terms of the intermediate representation.

## 3 Methodology

In open-world environments with real-time perception demands, models frequently encounter single-image data streams that contain OOD samples. To address this challenge, we propose *Open-set Intermediate-Representation-based Test-time Adaptation* (Open-IRT) in Fig. 2. Open-IRT is an open-set, single-image test-time adaptation strategy composed of two key modules: Polarity-aware Prompt based OOD Filter module and Intermediate Domain-based Test-time adaptation module.

### 3.1 PPF: Polarity-aware Prompt-based OOD Filtering Mechanism

As shown in Fig. 2a, we introduce the *Polarity-aware Prompt-based OOD Filter* (PPF) module, which improves ID–OOD separability by exploiting the dual-polarity semantics of vision-language models. Specifically, we design positive prompts $p_c$ (e.g., "a photo of a [CLS]") to represent ID prototypes, and negative prompts $p'_c$ (e.g., "a photo of no [CLS]") to encode inverse semantics [48]. Given a vision feature $f = g_v(x)$ after L2 normalization, we compute its cosine similarity with both the positive prompt $p_c$ and the negative prompt $p'_c$ for class $c$. $sim_{pos} = \text{sim}(f, p_c) = \frac{f^\top p_c}{\|f\|\|p_c\|}, sim_{neg} = \text{sim}(f, p'_c) = \frac{f^\top p'_c}{\|f\|\|p'_c\|}, sim_{diff} = |sim_{pos} - sim_{neg}|$.

**Hypothesis 1** (Semantic Contrast Hypothesis). *For ID samples $\forall c \in [C]$, $f \sim \mathcal{P}_I$ and OOD $f \sim \mathcal{P}_O$:*

$$\mathbb{E}_{f \sim \mathcal{P}_I}[\text{sim}(f, p_c)] \geq \tau_I, \quad \mathbb{E}_{f \sim \mathcal{P}_O}[\text{sim}(f, p_c)] \leq \tau_O \tag{1}$$

$$\mathbb{E}_{f \sim \mathcal{P}_I}[\text{sim}(f, p_c) - \text{sim}(f, p'_c)] \geq \Delta_I, \quad \mathbb{E}_{f \sim \mathcal{P}_O}[\text{sim}(f, p_c) - \text{sim}(f, p'_c)] \leq \Delta_O \tag{2}$$

*where thresholds satisfy $\tau_I > \tau_O$ and $\Delta_I > \Delta_O$.*

First, we propose the Semantic Contrast Hypothesis. For ID samples, the alignment with positive prompts is strong, i.e., $\text{sim}(f, p_c)$ is high, and the polarity gap $|\text{sim}(f, p_c) - \text{sim}(f, p'_c)|$ is also large, leading to a confidently larger overall sum. For OOD samples, the alignment with positive prompts is weaker, and the polarity gap is less marked due to semantic ambiguity, leading to suppressed overall sum. The proposed scoring function $\mathcal{S}(f)$ jointly captures *absolute semantic alignment* and

*relative semantic polarity* between an input and its paired prompts. This promotes effective ID–OOD separability by maximizing the distributional contrast between them.

$$\mathcal{S}(f) = \phi\left(\sup_{c\in[C]}\left[\text{sim}(f, p_c) + \alpha\left|\text{sim}(f, p_c) - \text{sim}(f, p_c')\right|\right]\right), \tag{3}$$

where $\phi$ is a min-max normalization operator to ensure the score lies within the range $[0, 1]$, and $\alpha$ controls the contrast intensity. As shown in Fig. 3, while individual sample have unique semantics, the resulting aggregate score distribution exhibits a clear bimodal pattern, demonstrating consistent statistical regularity across large-scale data. Furthermore, the introduction of the polarity gap term significantly enhances ID-OOD separation, thereby validating the Hypothesis 1. This dual-polarity design—leveraging alignment with positive prompts and contrast with negative prompts—offers an effective mechanism for ID-OOD distinction. Detailed theoretical analyses are provided in appendix.

Then, we introduce the score bank $\mathcal{B}^s$ and feature bank $\mathcal{B}^f$, both updated using a sliding window strategy, where it functions equivalently to a FIFO queue. The score bank $\mathcal{B}^s$ stores the scores of individual samples and is divided into two components: ID ($\mathcal{B}_I^s$) and OOD ($\mathcal{B}_O^s$) score banks, based on the Gaussian Mixture Model as described in Eq. 4.

$$\mathcal{P}(x) = \pi(x)\mathcal{N}(x \mid \mu_I^s, \sigma_I^{s\,2}) + (1 - \pi(x))\mathcal{N}(x \mid \mu_O^s, \sigma_O^{s\,2}) \tag{4}$$

Here, $\pi(x)$ denotes the probability that $S(x)$ belongs to the ID class, and $\mu_{I/O}^s$, $\sigma_{I/O}^{s\,2}$ are the mean and variance of the ID/OOD components. The probability $\pi(x)$ is computed using the Expectation-Maximization algorithm.

Upon the arrival of a new sample $x$, its score $S(x)$ is appended to both $\mathcal{B}_I^s$ and $\mathcal{B}_O^s$, and Eq. 4 is used to classify it as either ID ($\hat{b} = 1$) or OOD ($\hat{b} = 0$). To mitigate the impact of limited samples during the early stages of learning, we employ bootstrapped resampling, which helps prevent overfitting to sparse observations. The classified features are then inserted into the corresponding feature banks $\mathcal{B}_{I/O}^f$ for use in the subsequent module.

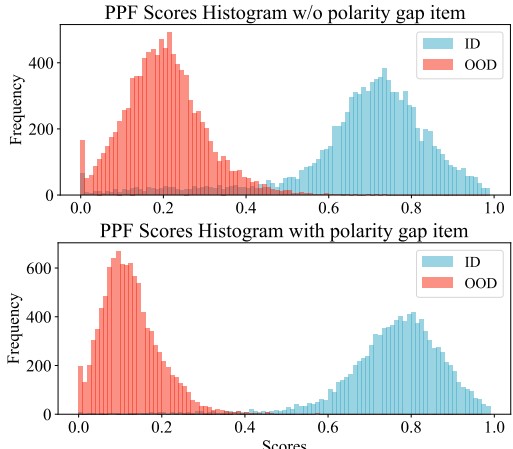

Figure 3: PPF score distribution histogram with CIFAR-10C [19] as ID and MNIST [22] as OOD.

### 3.2 IDT: Intermediate Domain-based Test-time adaptation

We now introduce the *Intermediate Domain-based Test-time adaptation* (IDT) module, which leverages the feature banks $\mathcal{B}_I^f$ and $\mathcal{B}_O^f$ to further refine adaptation. To address the challenge in Fig. 1, we first propose Hypothesis 2 that models the intermediate domain ($\mathcal{F}_M$).

**Hypothesis 2** (Intermediate Domain Characterization). *Let $\mathcal{F}_I \triangleq \{f | f \sim \mathcal{P}_I\}$ and $\mathcal{F}_O \triangleq \{f | f \sim \mathcal{P}_O\}$ denote the ID/OOD feature distributions respectively. There exists a measurable transformation $\mathcal{T} : \mathcal{F}_I \cup \mathcal{F}_O \to \mathcal{F}_M$ such that for the induced intermediate feature space $\mathcal{F}_M$ that satisfies the following approximate equality:*

$$d_{\mathcal{H}}(\mathcal{F}_I, \mathcal{F}_M) + d_{\mathcal{H}}(\mathcal{F}_O, \mathcal{F}_M) \approx d_{\mathcal{H}}(\mathcal{F}_I, \mathcal{F}_O), \tag{5}$$

*where $d_{\mathcal{H}}$ represents the Hilbert-Schmidt independence criterion-based dissimilarity.*

Rather than directly maximizing the ID-OOD distance ($\mathcal{F}_I, \mathcal{F}_O$), which is challenging due to limited knowledge of OOD distribution, we instead construct an intermediate domain $\mathcal{F}_M$ to act as a bridge. By encouraging both ID and OOD samples to move away from this intermediate domain (i.e., increasing $d_{\mathcal{H}}(\mathcal{F}_I, \mathcal{F}_M)$ and $d_{\mathcal{H}}(\mathcal{F}_O, \mathcal{F}_M)$), we indirectly enhance the overall discrepancy $d_{\mathcal{H}}(\mathcal{F}_I, \mathcal{F}_O)$. This aligns with the intuition of margin-based separation in representation space.

**Contrastive Learning-based Middle Domain Loss.** We first model the intermediate domain feature $f_m$ as per Hypothesis 2. To normalize the sample features $f_s$, we compute their mean and variance

based on augmented features, which is derived the $f_s$ as Eq. 6. The method for calculating $f_s$ has been validated for its rationality in [62], where it is computed across spatial dimensions independently for each feature channel. The feasibility of this approach lies in preserving channel-specific style information. The variance for each channel emphasizes fine-grained structural characteristics in that feature dimension, such as color, texture intensity, etc., without mixing with other channels, thereby achieving refined style representation and alignment.

$$f_s = \frac{f - \mu(f)}{\sigma(f)} \tag{6}$$

Then, we extract the style features stored in the feature memory banks $\mathcal{B}^f_{I/O}$ to compute $\mu^f_{I/O}$ and $\sigma^f_{I/O}$. The design of intermediate style feature $f_m$ is theoretically motivated by domain adaptation principles [62], which suggest that constructing an intermediate feature space through style-transfer-like transformations can effectively position the new domain between source and target domains, enabling more controlled feature interpolation. $f_m$ is then re-assigned as follows.

$$f_m = \begin{cases} f_s \cdot \sigma^f_O + \mu^f_O & \text{if } \hat{b} = 1; \text{Conf}(x) > \mu_I \\ f_s \cdot \sigma^f_I + \mu^f_I & \text{if } \hat{b} = 0; \text{Conf}(x) < \mu_O \end{cases} \tag{7}$$

The confidence Conf(x) is defined in Eq. 11. To further enhance inter-class discriminability and suppress intra-class noise, we introduce two thresholds, $\mu_I$ and $\mu_O$. They represent the mean confidence scores of the accumulated 512 ID/OOD Conf($x$), respectively, updated by a sliding window mechanism. If Conf($x$) > $\mu_I$, we treat the input as a confident ID sample. If Conf(x) < $\mu_O$, it is more likely to be a noisy sample or an OOD sample6. The effectiveness of $f_m$ construction is visually confirmed in Fig. 4, where the T-SNE visualization shows a clear separation between ID

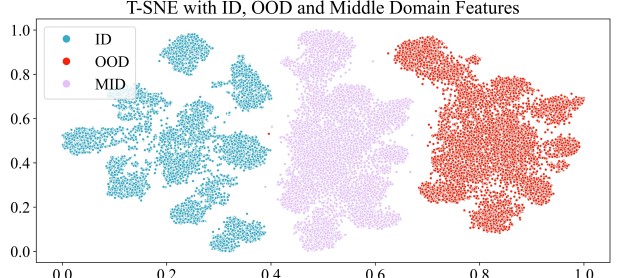

Figure 4: T-SNE visualization of MNIST [22] (ID) and CIFAR-10C [19] (OOD) and their Middle-Domain (MID)

and OOD features, with a distinct intermediate cluster. This indicates that the constructed intermediate representations indeed satisfy the desired relational constraints in the feature space, thereby validating the theoretical motivation of Hypothesis 2. Then, we select the $K$ nearest neighbors $\mathcal{N}_I$ and $\mathcal{N}_O$, for the sample feature $f$ from the feature banks $\mathcal{B}^f_I$ and $\mathcal{B}^f_O$, respectively. When the sample is confidently distinguished as ID ($\hat{b} = 1$ and Conf($x$) > $\mu_I$), the loss $\mathcal{L}_I$ based on InfoNCE loss [63] is constructed as follows.

$$\mathcal{L}_I = -\frac{1}{K^+} \sum_{z^+ \in \mathcal{N}_I} (\log \frac{\exp(\text{sim}(f, z^+)/\tau)}{\sum_{z^- \in \mathcal{N}_O} \exp(\text{sim}(f, z^-)/\tau)}$$
$$-\lambda \log \frac{\exp(\text{sim}(f_m, z^+)/\tau)}{\sum_{z^- \in \mathcal{N}_O} \exp(\text{sim}(f_m, z^-)/\tau)}) \mathbb{1}(y^+ = \hat{y}_p) \tag{8}$$

where $K^+ = \sum_{z^+ \in \mathcal{N}_I} \mathbb{1}(y^+ = \hat{y}_p)$ denotes the number of positively matched neighbors with the pseudo label $\hat{y}_p$. For confident ID samples, the objective of $\mathcal{L}_I$ is to align with ID while minimizing similarity to OOD and distancing from the intermediate feature $f_m$.

$$\mathcal{L}_O = -\frac{1}{K} \sum_{z^+ \in \mathcal{N}_O} \left( \log \frac{\exp(\text{sim}(f, z^+)/\tau)}{\sum_{z^- \in \mathcal{N}_I} \exp(\text{sim}(f, z^-)/\tau)} - \lambda \log \frac{\exp(\text{sim}(f_m, z^+)/\tau)}{\sum_{z^- \in \mathcal{N}_I} \exp(\text{sim}(f_m, z^-)/\tau)} \right) \tag{9}$$

For confident OOD samples ($\hat{b} = 0$ and Conf($x$) < $\mu_O$), the objective of $\mathcal{L}_O$ is to improve sensitivity to OOD samples by pulling OOD features away from both ID and the intermediate domain feature.

The differences between Eq. 8 and Eq. 9 arise from the distinction between ID and OOD categories in the standard OOD detection task. For ID data, the model needs to learn fine-grained semantic consistency and achieve precision for each class, which is why we set $\mathbb{1}(y^+ = \hat{y}_p)$ in Eq. 8.

$$\mathcal{L}_{mid} = \begin{cases} \mathcal{L}_I & \text{if } \hat{b} = 1; \text{ Conf}(x) > \mu_I \\ \mathcal{L}_O & \text{if } \hat{b} = 0; \text{ Conf}(x) < \mu_O \\ 0 & \text{otherwise} \end{cases} \tag{10}$$

**Threshold-based Bidirectional Pseudo Loss.** To facilitate positive knowledge while mitigating intra-class noise, we design an effective pseudo-label strategy. First, we calculate the confidence $\text{Conf}(x)$ using the maximum cosine similarity between feature $g_v(x)$ and classifier weights $\mathbf{C}$:

$$\text{Conf}(x) = \max_i(g_v(x) \cdot \mathbf{C}^T), \quad \hat{y}_p = \arg\max_i(g_v(x) \cdot \mathbf{C}^T) \tag{11}$$

We analyze $\text{Conf}(x)$ from two perspectives: For high-confidence scenarios, we align predictions with $\hat{y}_p$. For low-confidence scenarios, $\hat{y}_p$ may introduce noise, so we stop relying on it and use reverse optimization to push predictions away from it, mitigating the risk of misclassification.

$$\mathcal{L}_{CE}(x, \hat{y}) = -\frac{1}{N} \sum_{i=1}^{N} (\hat{y}_i \log(p_i) + (1 - \hat{y}_i) \log(1 - p_i)) \tag{12}$$

If $\text{Conf}(x) > \mu_I$, we treat the input as an confident sample and optimize the model to align predictions with $\hat{y}_p$ using $\mathcal{L}_{CE}$, enhancing the inter-class discriminability. If $\text{Conf}(x) < \mu_O$, we deviate the prediction results from the low-quality pseudo-label $\hat{y}_p$ by $-\mathcal{L}_{CE}$, suppressing the intra-class noise.

$$\mathcal{L}_{psd} = \begin{cases} \mathcal{L}_{CE}(x, \hat{y}_p) & \text{if } \hat{b} = 1; \text{ Conf}(x) > \mu_I \\ -\mathcal{L}_{CE}(x, \hat{y}_p) & \text{if } \hat{b} = 0; \text{ Conf}(x) < \mu_O \\ 0 & \text{otherwise} \end{cases} \tag{13}$$

**Total Test-time Adaptation Loss.** The total loss function $\mathcal{L}_{TTA}$ combines the pseudo-label loss $\mathcal{L}_{psd}$ and the intermediate domain loss $\mathcal{L}_{mid}$. $\mathcal{L}_{psd}$ aims to enhance inter-class discriminability and suppress intra-class noise, while $\mathcal{L}_{mid}$ indirectly enlarge the ID-OOD distance by increasing both $d_{\mathcal{H}}(\mathcal{F}_I, \mathcal{F}_M)$ and $d_{\mathcal{H}}(\mathcal{F}_O, \mathcal{F}_M)$ in Hypothesis 2. This combination enables dynamic adjustment of the model's adaptation strategy.

$$\mathcal{L}_{TTA} = \mathcal{L}_{psd} + \mathcal{L}_{mid} \tag{14}$$

During test time, each incoming sample is used to update the model parameters via backpropagation with the test-time adaptation loss $\mathcal{L}_{TTA}$. Subsequently, the same sample is used to compute the evaluation metrics. This online update-then-evaluate approach simulates a realistic scenario where the model continuously adapts to distributional shifts without access to ground-truth labels.

## 4 Experiments

### 4.1 Implementation Details

**Datasets.** We utilize representative ID and OOD datasets to ensure a complete evaluation. For ID datasets, we leverage CIFAR-10C/100C [19], ImageNet-C [19], and VisDA [21]. The OOD datasets include MNIST [22], MNIST-M [23], SVHN [24], Tiny-ImageNet [20], and CIFAR-10C/100C [19].

**Metrics.** We utilize the Area Under the Receiver Operating Characteristic Curve (AUROC), the False Positive Rate at 95% True Positive Rate (FPR95) and $Acc_{HM}$ as main metrics. Here, $Acc_{HM}$ denotes the harmonic mean of $Acc_I$ and $Acc_O$, where $Acc_O$ is the precision of ID-OOD binary classification, and $Acc_I$ is the general accuracy to correctly identify ID categoriess.

**Baselines.** We utilize the CLIP [10] and MaPLe [11] as models, which are based on the ViT-B16/ViT-B32 [64] architectures. We adopt ZS-Eval [52], TPT/TPT-C [12], PAlign/PAlign-C [46], TDA [61], DPE [65], UniEnt [58], OWTTT [66] and ROSITA [52] as baselines.

**Details.** All experiments are reproduced based on publicly available code, with ImageNet-C experiments conducted on an NVIDIA A6000 GPU and all other experiments on an NVIDIA 3090 GPU. In main experiments, the test size for both ID and OOD datasets is 10,000, except for the VisDA in

Table 3, which is 50,000. In Table 5, the OOD test size is $ratio \times 10,000$. The text encoder is kept fixed, while the vision encoder is updated with a SGD optimizer with learning rate of 1.5e-3, and batch size is set to 1 for all experiments. The size $B$ of both score bank $\mathcal{B}^s$ and feature bank $\mathcal{B}^f$ are set to 128, the number of nearest neighbors $K$ in Eq. 8, 9 configured to 5. The $\alpha$ in Eq. 3 is set to 0.2, and $\lambda$ in Eq. 8, 9 are set to 0.1, respectively. Details on the baselines and analysis of hyper-parameters are provided in appendix.

| | Method | MNIST | | | SVHN | | | Tiny-ImageNet | | | CIFAR-100C | | |
|---|---|---|---|---|---|---|---|---|---|---|---|---|---|
| | | AUC ↑ | FPR ↓ | HM ↑ | AUC ↑ | FPR ↓ | HM ↑ | AUC ↑ | FPR ↓ | HM ↑ | AUC ↑ | FPR ↓ | HM ↑ |
| CIFAR10-C / CLIP | ZS-Eval [52] | 91.91 | 85.22 | 75.60 | 89.94 | 64.25 | 74.11 | 91.33 | 27.13 | 74.24 | 82.57 | 67.96 | 68.92 |
| | TPT [12] | 91.90 | 85.70 | 75.78 | 89.93 | 64.54 | 74.30 | 91.31 | 27.26 | 74.98 | 82.57 | 68.09 | 69.13 |
| | TPT-C [12] | 83.21 | 67.03 | 75.05 | 60.83 | 69.47 | 50.63 | 74.12 | 57.34 | 48.88 | 63.76 | 93.05 | 51.98 |
| | ROSITA [52] | 99.43 | 3.25 | 83.95 | 94.94 | 31.22 | 79.12 | 96.37 | 12.69 | 80.07 | 83.01 | 64.54 | 69.64 |
| | OWTTT [66] | 98.05 | 12.50 | 83.27 | 80.74 | 50.33 | 70.10 | 87.09 | 52.29 | 73.98 | 62.55 | 91.68 | 56.46 |
| | TDA [61] | 92.94 | 71.11 | 77.06 | 92.02 | 52.68 | 76.64 | 91.68 | 25.37 | 75.94 | 83.54 | 66.06 | 70.13 |
| | UniEnt [58] | 91.98 | 85.20 | 75.62 | 89.97 | 64.38 | 74.18 | 91.40 | 26.96 | 74.73 | 82.59 | 68.14 | 68.98 |
| | DPE [65] | 46.97 | 99.10 | 27.60 | 84.15 | 85.24 | 68.52 | 89.92 | 31.30 | 69.90 | 79.18 | 75.06 | 62.34 |
| | Open-IRT | 99.73 | 1.28 | 84.55 | 96.52 | 18.34 | 80.62 | 97.07 | 10.09 | 80.95 | 82.65 | 61.69 | 69.20 |
| MAPLE | ZS-Eval [52] | 98.16 | 5.50 | 82.43 | 98.35 | 7.82 | 83.58 | 90.86 | 27.53 | 76.01 | 86.15 | 52.00 | 71.68 |
| | TPT [12] | 98.16 | 69.35 | 81.74 | 98.34 | 7.88 | 82.67 | 90.86 | 27.55 | 75.40 | 86.15 | 52.10 | 70.84 |
| | TPT-C [12] | 98.22 | 5.15 | 83.34 | 98.35 | 7.85 | 83.55 | 90.91 | 27.44 | 75.84 | 86.20 | 51.96 | 71.60 |
| | PAlign [46] | 98.16 | 5.62 | 82.57 | 98.34 | 7.88 | 83.44 | 90.86 | 27.55 | 76.03 | 86.15 | 52.10 | 71.50 |
| | PAlign-C [46] | 98.61 | 3.45 | 83.91 | 98.35 | 8.13 | 83.45 | 91.17 | 26.95 | 76.12 | 86.53 | 50.64 | 71.11 |
| | ROSITA [52] | 99.45 | 3.84 | 87.71 | 98.02 | 11.45 | 84.56 | 91.76 | 25.23 | 77.60 | 86.92 | 48.12 | 72.79 |
| | OWTTT [66] | 98.34 | 9.63 | 86.52 | 71.01 | 78.78 | 68.70 | 71.20 | 85.81 | 68.29 | 62.35 | 88.44 | 61.89 |
| | TDA [61] | 98.42 | 4.13 | 81.97 | 98.60 | 6.20 | 83.95 | 91.27 | 27.00 | 76.84 | 86.72 | 51.40 | 72.61 |
| | UniEnt [58] | 98.17 | 5.49 | 82.64 | 98.35 | 7.85 | 83.65 | 90.90 | 27.41 | 76.08 | 86.16 | 51.91 | 71.72 |
| | DPE [65] | 83.82 | 92.73 | 55.52 | 97.42 | 12.95 | 79.41 | 89.10 | 31.13 | 74.32 | 73.57 | 73.67 | 53.64 |
| | Open-IRT | 99.51 | 2.85 | 88.11 | 97.62 | 15.92 | 85.01 | 91.83 | 24.38 | 77.80 | 87.42 | 46.40 | 73.20 |

Table 1: Open-set Single-Image Test-time adaptation results with CIFAR-10C as ID, MNIST, SVHN, Tiny-ImageNet, and CIFAR-100C as OOD. The metrics include AUROC (AUC), FPR95 (FPR), and $Acc_{HM}$ (HM) as defined in Section 4.1. Results in bold represent the best performance, while underlined results indicate the second-best ones.

## 4.2 Main Result

**Cifar Benchmark.** The cifar benchmark leverages CIFAR-10C/100C [19] as ID datasets, and MNIST [22], SVHN [24], and Tiny-ImageNet [20] as OOD datasets. CIFAR-100C/10C is also treated as an OOD case with small distribution shifts, applying label offsets to distinguish from ID data. As shown in Table 1, Open-IRT has demonstrated significant performance advantages. For example, in the CIFAR-10C → SVHN case with CLIP, Open-IRT achieved notable improvements by +1.58% AUROC, -12.88% FPR95, and +1.50% $Acc_{HM}$. Additionally, for the CIFAR-100C → SVHN case with CLIP, Open-IRT achieves an +8.45% in $Acc_{HM}$, -10.80% in FPR95, and +11.04% in AUROC, as detailed in appendix. OpenIRT also achieves better results in MaPLe, a multi-modal fine-tuning framework for CLIP, due to its adaptability across fine-tuning methods through direct feature space operation. Furthermore, Open-IRT exhibits obvious improvements in the FPR95 (up to -12.88%), highlighting its effectiveness in reducing the misclassification of OOD samples. This can be attributed to Open-IRT's ability to model the ID-OOD distribution effectively (see Fig. 3).

| | CLIP | | | | | | MAPLE | | | | | |
|---|---|---|---|---|---|---|---|---|---|---|---|---|
| Method | IN-C→MNIST | | | IN-C→SVHN | | | IN-C→MNIST | | | IN-C→SVHN | | |
| | AUC ↑ | FPR ↓ | HM ↑ | AUC ↑ | FPR ↓ | HM ↑ | AUC ↑ | FPR ↓ | HM ↑ | AUC ↑ | FPR ↓ | HM ↑ |
| ZS-Eval [52] | 93.34 | 57.34 | 41.41 | 85.72 | 74.34 | 40.84 | 81.24 | 93.97 | 41.29 | 83.05 | 73.63 | 42.44 |
| TPT [12] | 91.89 | 59.54 | 41.02 | 85.03 | 48.98 | 40.16 | 80.31 | 93.54 | 39.13 | 82.67 | 73.57 | 39.90 |
| TPT-C [12] | 57.84 | 98.92 | 6.37 | 10.31 | 99.59 | 7.29 | 82.88 | 87.95 | 41.13 | 82.17 | 72.10 | 41.37 |
| OWTTT [66] | 95.76 | 10.43 | 42.95 | 87.75 | 26.23 | 38.50 | 98.58 | 3.35 | 48.69 | 77.17 | 39.74 | 38.10 |
| TDA [61] | 90.54 | 76.23 | 43.66 | 86.76 | 75.45 | 43.07 | 76.76 | 99.02 | 42.98 | 82.46 | 91.75 | 44.63 |
| UniEnt [58] | 94.19 | 46.98 | 41.53 | 87.56 | 67.03 | 41.10 | 81.53 | 93.45 | 41.50 | 83.41 | 70.84 | 42.78 |
| DPE [65] | 87.92 | 91.94 | 42.87 | 82.96 | 77.90 | 41.93 | 73.97 | 99.59 | 41.39 | 80.06 | 87.10 | 44.05 |
| ROSITA [52] | 98.97 | 8.55 | 45.74 | 91.90 | 45.66 | 38.86 | 97.19 | 9.56 | 48.28 | 91.86 | 29.21 | 44.47 |
| Open-IRT | 99.44 | 1.06 | 49.49 | 98.45 | 9.37 | 48.19 | 97.54 | 10.97 | 50.07 | 95.54 | 23.11 | 48.77 |

Table 2: Open-set Single-Image Test-time adaptation. The ID data is ImageNet-C, while the OOD data comprises MNIST and SVHN.

| $\mathcal{L}_{psd}$ | $\mathcal{L}_{mid}$ | CIFAR-10C→MNIST | | | CIFAR-10C→SVHN | | | VisDA→MNIST | | | VisDA→SVHN | | |
|---|---|---|---|---|---|---|---|---|---|---|---|---|---|
| | | AUC↑ | FPR↓ | HM↑ | AUC↑ | FPR↓ | HM↑ | AUC↑ | FPR↓ | HM↑ | AUC↑ | FPR↓ | HM↑ |
| | ✗ | ✗ | 91.91 | 84.91 | 78.07 | 89.94 | 64.03 | 75.14 | 93.55 | 65.83 | 84.34 | 90.45 | 64.98 | 79.02 |
| | ✗ | ✓ | 99.60 | 1.90 | 78.91 | 94.58 | 32.42 | 75.46 | 99.62 | 2.51 | 89.73 | 99.10 | 5.35 | 89.29 |
| CLIP | ✓ | ✗ | 98.15 | 1.74 | **85.52** | 93.52 | 37.04 | 78.57 | 95.66 | 35.40 | 87.74 | 95.04 | 33.70 | 85.67 |
| | ✓ | $\lambda=0$ | 99.70 | 1.90 | 83.52 | 95.92 | 24.29 | 79.24 | 99.79 | 1.53 | 89.83 | 97.58 | 4.26 | 89.81 |
| | ✓ | ✓ | **99.71** | **1.28** | 84.52 | **96.52** | **18.35** | **80.61** | **99.84** | **1.27** | **90.85** | **99.20** | **3.05** | **90.39** |

Table 4: Ablation Experiments. The ID data is a combination of CIFAR-10C and VisDA, while the OOD data comprises MNIST and SVHN.

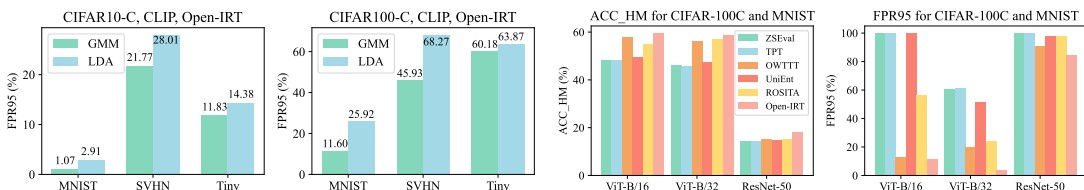

(a) Analyze of Open-IRT with GMM and LDA.

(b) Backbone ViT-B/16, ViT-B/32 and ResNet50.

Figure 5: Comparison of GMM and LDA in PPF and different backbone architectures.

**VisDA and ImageNet Benchmarks.** The VisDA [21] and ImageNet-C [19] serve as ID datasets, while MNIST [22], MNIST-M [23], and SVHN [24] serve as OOD datasets. The complexity of VisDA and ImageNet-C introduces additional challenges for OOD detection. In Table 2, Open-IRT achieves notable improvements over existing baselines. For example, in ImageNet-C→MNIST experiment with CLIP, Open-IRT achieves gains of +0.47% AUROC, -7.49% FPR95, and +3.75% $Acc_{HM}$. Similarly, in Table 3, Open-IRT achieves gains of +1.58% AUROC, -7.46% FPR95, and +1.09% $Acc_{HM}$ in VisDA → MNIST-M with CLIP. Moreover, compared to Table 1, when the ID dataset becomes more complex (e.g., ImageNet-C), the model's performance in OOD detection tasks does not fully saturate, resulting in a relatively lower $Acc_{HM}$. In such cases, Open-IRT's improvement in $Acc_{HM}$ is more obvious, such as a +5.12% increase in the ImageNet-C → SVHN experiment with CLIP. Additionally, Open-IRT achieves gains of -16.86% FPR95 and +6.55% AUROC. Since Open-IRT operates primarily in the feature space and enhances the separation of the ID-OOD boundary (see Fig. 4), its effectiveness becomes increasingly apparent as the feature distribution grows more complex.

| | Method | VisDA→MNIST | | | VisDA→MNIST-M | | |
|---|---|---|---|---|---|---|---|
| | | AUC↑ | FPR↓ | HM↑ | AUC↑ | FPR↓ | HM↑ |
| CLIP | ZS-Eval [52] | 93.55 | 65.86 | 78.30 | 87.25 | 67.10 | 74.84 |
| | TPT [12] | 93.55 | 66.11 | 78.44 | 87.25 | 67.19 | 75.05 |
| | TPT-C [12] | 81.81 | 85.12 | 75.09 | 87.44 | 62.31 | 77.32 |
| | ROSITA [52] | 99.63 | 2.99 | 90.59 | 97.10 | 15.14 | 86.88 |
| | Open-IRT | **99.85** | **1.27** | **90.88** | **98.68** | **7.68** | **87.97** |
| MAPLE | ZS-Eval [52] | 93.07 | 66.13 | 80.29 | 92.31 | 45.66 | 78.83 |
| | TPT [12] | 93.07 | 66.03 | 80.35 | 92.30 | 45.70 | 78.87 |
| | TPT-C [12] | 93.40 | 59.35 | 80.35 | 92.48 | 44.17 | 78.93 |
| | PAlign [46] | 93.07 | 66.03 | 80.62 | 92.29 | 45.70 | 79.17 |
| | PAlign-C [46] | 95.61 | 27.65 | 81.93 | 94.13 | 32.97 | 81.48 |
| | ROSITA [52] | 99.80 | 1.40 | **90.84** | 98.90 | 5.79 | 89.40 |
| | Open-IRT | **99.87** | **1.01** | 90.82 | **99.15** | **4.99** | **89.56** |

Table 3: VisDA (ID), MNIST/MNIST-M (OOD) Results.

### 4.3 Experiment Analysis

**Score Ablation in PPF.** We assess the effectiveness of our PPF mechanism in Fig. 5a, comparing PPF with GMM and LDA [67]. The results indicate that the PPF with GMM performs better results, such as -2.55% FPR95 improvement in CIFAR-10C → Tiny-ImageNet task. This performance gain can be attributed to GMM's ability to model complex and non-linearly separable feature distributions, which more accurately capture the characteristics of real-world domain shifts. In contrast, LDA relies on the assumption of linear decision boundaries, which limits its capacity to handle such complexities.

**Loss Ablation in IDT.** We analyze the effectiveness of each component of IDT in Eq.14, as shown in Table 4, with the first row representing the zero-shot evaluation scenario. The results demonstrate that $\mathcal{L}_{mid}$ significantly improves FPR95 and AUROC, enhancing ID-OOD binary classification by increasing the separation between ID and OOD samples in feature space. On complex datasets like VisDA, $\mathcal{L}_{mid}$ outperforms $\mathcal{L}_{psd}$ by +1.99% and +3.62% in $Acc_{HM}$, respectively. However, on simpler datasets like CIFAR-10C, $\mathcal{L}psd$ performs better in $Acc_{HM}$ as it helps reduce intra-class noise, which is more beneficial for simpler features. Furthermore, setting $\lambda = 0$ to evaluate the intermediate-domain strategy shows that introducing the intermediate-domain improves performance by up to -5.94% in FPR95, supporting our Hypothesis 2.

**Analyze Backbones.** In Fig. 5b, we utilize ViT-B/16, ViT-B/32 and ResNet50 as the backbones in CIFAR-100C → MNIST task with CLIP. The results demonstrate that Open-IRT achieves the highest $Acc_{HM}$ (e.g., 59.56% in ViT-B/16, 58.62% in ViT-B/32) and the lowest FPR95 (e.g., 11.29% in ViT-B/16, 3.74% in ViT-B/32). These results highlight the robustness of Open-IRT, as it directly optimizes the feature space, independent of the backbone type, demonstrating its broad applicability.

**Varying OOD Ratios.** To examine the robustness of Open-IRT, we conduct CIFAR-10C → MNIST experiments on CLIP under different OOD ratios with additional experiments are in appendix. As shown in Table 5, Open-IRT consistently achieves better results across all OOD ratios with $Acc_{HM}$ fluctuate by only 1.01%. This stability suggests that Open-IRT is better suited to handle the uncertainties associated with fluctuating OOD proportions.

| | ratio | 0.2 | 0.4 | 0.6 | 0.8 | 1.0 |
|---|---|---|---|---|---|---|
| CLIP | ZS-Eval [52] | 75.55 | 75.60 | 75.59 | 75.59 | 75.60 |
| | TPT [12] | 75.77 | 75.78 | 75.81 | 75.76 | 75.78 |
| | TPT-C [12] | 73.05 | 74.29 | 74.75 | 75.05 | 75.05 |
| | DPE [65] | 65.67 | 66.12 | 56.38 | 29.98 | 27.60 |
| | OWTTT [66] | 62.31 | 68.85 | 81.70 | 82.90 | 83.27 |
| | TDA [61] | 72.45 | 75.04 | 77.54 | 77.91 | 77.06 |
| | ROSITA [52] | 83.21 | 84.68 | 83.90 | 83.89 | 83.95 |
| | Open-IRT | **85.10** | **85.53** | **85.08** | **84.63** | **84.52** |

Table 5: Experiments ($Acc_{HM}$) on CLIP with different OOD ratios with CIFAR-10C (ID), and MNIST (OOD).

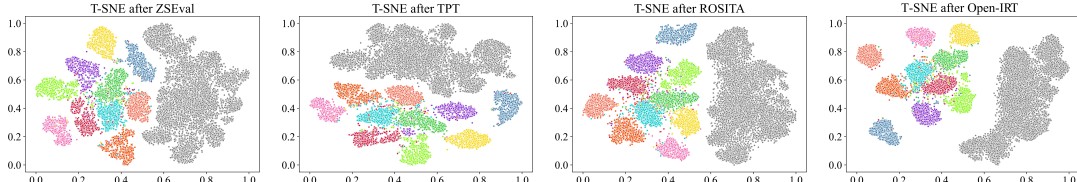

Figure 6: T-SNE Visualization with gray points as OOD.

**T-SNE Visualization.** We conduct T-SNE experiments of CIFAR-10C → MNIST. The T-SNE visualization in Fig. 6 reveals that Open-IRT achieves better separation and more compact clustering within the ID classes, while also establishing a clearer boundary between ID and OOD classes.

## 5 Conclusion

In this paper, we address the challenges of real-time adaptation in open-set environments with single-sample stream and propose the *Open-set Intermediate-Representation-based Test-time adaptation* (Open-IRT) framework. Its Polarity-aware Prompt-based OOD Filter module leverages the rich cross-modal information of vision language models, corporating both absolute semantic alignment and relative semantic polarity. The Intermediate Domain-based Test-time adaptation module constructs an intermediate domain and indirectly decomposes and enlarge the ID-OOD distributional discrepancy in real-time. Experiments across various benchmarks underscores the Open-IRT's potential to enhance the robustness and adaptability in dynamic real-world. Future work can focus on extending its application to real-world object detection or semantic segmentation. Moreover, a tension exists between the method's requirement for a sizable memory bank and its premise of single-image test-time adaptation. Future work should aim to reconcile this, achieving robust performance under a strictly single-image setting.

**Acknowledgements**: This work was supported by the National Key Research and Development Program of China (No. 2024YFE0211000), in part by the National Natural Science Foundation of China (No. 62372329), in part by the Shanghai Scientific Innovation Foundation (No. 23DZ1203400), in part by the China Postdoctoral Science Foundation (No. BX20250383, GZB20250385, 2025M771530, 2025M771539), in part by Tongji-Qomolo Autonomous Driving Commercial Vehicle Joint Lab Project, and in part by Xiaomi Young Talents Program.

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
