# OpenReview forum: "OOD-Barrier: Build a Middle-Barrier for Open-Set Single-Image Test Time Adaptation via Vision Language Models"
_NeurIPS.cc/2025/Conference — NeurIPS 2025 poster_

### Official Review · Reviewer_g3am · 2025-06-19

**Clarity:** 2
**Significance:** 3
**Originality:** 3
**Rating:** 4
**Confidence:** 5

**Summary:**

This paper introduces Open-IRT, a single-image open-set Test-Time Adaptation framework for VLMs. The method proposes to use semantical text understanding to build a statistical barrier between ID and OOD samples. The method comprises different procedures, including preserving a memory of scores and a memory of features, the introduction of an intermediate domain based on confidence thresholds, and the employment of InfoNCE to contrast each image with respect to this intermediate domain. The method is evaluated on a series of popular TTA benchmarks, and compared to previous adaptation methods applied on CLIP and MaPLe, showing competitive results in harmonic accuracy and OOD detection metrics.

**Questions:**

1. Harmonic accuracy is used to cover both ID accuracy and detection accuracy, but what is the traditional ID accuracy for the experiments). Open-set TTA is a combination of two problems: a) OOD detection, and b) ID classification. This harmonic accuracy does not allow us to see at which of both does Open-IRT excel at, or if it is actually competitive in both.
2. How can we be sure that the good detection performance (problem a)) is not due to the fact that the ID images are corrupted (or synthetic in the case of VisDA-C), and the OOD images are not? Using corrupted versions of the OOD datasets would be beneficial. In the opposite sense, what happens natural images are used as ID datasets? Think of Imagenet, CIFAR-10, CIFAR-100, etc.
3. Is the final ID/OOD score the one computed from Eq. 3? This part is not very clear. As I mentioned before, although the writing is clear zooming in, the narrative can get confused when we zoom out Section 3.
4. What's the performance of other closed-set methods? TENT for instance, can be seamlessly applied on VLMs by just doing entropy minimization on the softmax cosine similarity logits. Moreover, other recent methods have been proposed: WATT(Neurips '24, model ensembling), OSTTA (ICCV '23, confidence-based detection), which are not included.

**Ethical Concerns:**

["NO or VERY MINOR ethics concerns only"]

**Final Justification:**

The paper provides a new method to deal with OOD TTA, a field that is just recently gaining traction in the context of VLMs. The method is sound, and the results demonstrate to be competitive. A final version of this paper would require adding additional results as the ones mentioned in my review, as well as improving the storytelling of the method for clarity, as it is charged with many concepts that need to be processed by the reader. I suggest a borderline accept and leave the final decision to the judgment of the Area Chair.

**Limitations:**

Yes

**Paper Formatting Concerns:**

No formatting issues

**Quality:**

2

**Strengths And Weaknesses:**

Strenghts:

-The ideas are novel and tackle important issues in the field of (open-set) TTA: single image adaptation, modelling of the ID-OOD distributions without using batches of images, and explicitly enforcing separability between ID and OOD samples in the feature space (which is not often done, with exceptions like UniEnt).
-Experiments include different popular benchmarks and a significant number of baselines on two different VLMs, showing competitive results.
-Using InfoNCE to create and contrast an intermediate domain is sound and clever.

Weaknesses:

-The paper is well written and generally clear, but information is very condensed and feels crowded by the end of Section 3; a better narrative could be possible.
-Most of the most recent open-set TTA methods are evaluated against Open-IRT, however, the performance of other closed-set strategies is overlooked, specially in VLMs. For instance, methods such as WATT, TENT or OSTTA (pioneer in open-set) could be present for completeness.
-Although several benchmarks are included, using the standard Imagenet (or any other representative of natural images) would further demonstrate to what extent this method works because of its open-set detection capabilities, an not only due to the fact that some images are corrupted and the OOD datasets are not. Similarly, having corrupted versions of MNIS, SVHN, etc., would aid to the completness of the study.

---

> ### Author Rebuttal · Authors · 2025-07-29
>
> ### **We sincerely thank you for your time and effort in reviewing our work. We are grateful for your recognition of the novelty of our idea and the robustness of our methodology.**
>
> Thank you for your feedback on the experimental section. We believe that your concerns can be addressed through a review of the original details, as well as the additional experiments and analysis provided. To facilitate your reading and help you locate **key points** in the original text, we have highlighted the important parts in **bold**.
>
> ---
>
> ### **\[Weakness 1] Condensed Information in Paper**
> We sincerely appreciate your recognition of the detailed aspects of our work, as well as your constructive suggestions regarding the overall structure. In our writing, we aimed to convey as much relevant information as possible to facilitate understanding, which may have led to an overly condensed presentation. In future revisions, we will streamline certain parts of the text and reallocate some of the detailed content to the supplementary material for improved clarity and readability.
>
> ---
>
> ### **\[Weakness 2] Additional Baselines (Tent \[2], WATT \[3], OSTTA \[1])**
>
> Thank you for your suggestion. Our work specifically targets **single-image-level test-time adaptation (TTA)** in **open-set recognition**. Many classical TTA methods, including Tent, WATT, and OSTTA, were not originally designed for this specific setting. To ensure a fair comparison, we adapted these methods accordingly (Line 660) to align with our evaluation protocol.
>
> Due to space limitations, we have to adopt representative baselines in the main paper. In response to your comments, we have conducted additional experiments incorporating Tent, WATT, and OSTTA. Specifically:
>
> 1. For the classical closed-set TTA method **Tent**, we integrated it with our original Zero-shot Evaluation (ZS-Eval) setup.
> 2. For **WATT** and **OSTTA**, we applied a fair and consistent evaluation strategy as described in Line 660.
>
> | Method (ViT‑B32) |         | MNIST             |        |        | SVHN              |        |        | Tiny‑ImageNet       |        |        |
> |:----------------:|:-------:|:-----------------:|:------:|:------:|:-------------------:|:------:|:------:|:-------------------:|:------:|:------:|
> |                  |         | AUC  ↑            | FPR ↓  | HM ↑   | AUC  ↑            | FPR ↓  | HM ↑   | AUC ↑               | FPR ↓  | HM ↑   |
> | CIFAR‑10         | ZS‑Eval | 96.74             | 18.87  | 73.28  | 92.12             | 43.98  | 71.37  | 91.59               | 24.66  | 72.28  |
> |                  | Tent [2] | 96.88            | 17.95  | 73.84  | 92.24             | 42.70  | 71.91  | 91.72               | 23.90  | 72.64  |
> |                  | TPT [4]    | 96.51             | 19.49  | 73.88  | 92.24             | 44.36  | 71.89  | 91.56               | 24.78  | 73.66  |
> |                  | WATT [3] | 96.59            | 18.98  | 74.30  | 92.18             | 43.79  | 72.20  | 91.58               | 24.33  | 73.79  |
> |                  | OSTTA [1] | 97.21           | 14.53  | 76.34  | 92.63             | 41.87  | 73.00  | 91.91               | 23.27  | 75.40  |
> |                  | ROSITA [5]  | 99.18             | 3.82   | 81.72  | 93.39             | 33.47  | 74.59  | 98.88               | 4.12   | 80.93  |
> |                  | Open‑IRT | **99.57**         | **1.66** | **82.35** | **93.95**     | **32.19** | **75.78** | **99.11**           | **3.56** | **81.57** |
> | CIFAR‑100        | ZS‑Eval | 89.27             | 60.99  | 46.18  | 78.25             | 79.92  | 44.64  | 72.66               | 61.23  | 45.77  |
> |                  | Tent [2] | 89.42            | 59.81  | 46.53  | 78.44             | 78.67  | 44.87  | 72.85               | 60.32  | 45.84  |
> |                  | TPT [4]     | 89.39             | 60.27  | 46.12  | 78.25             | 80.14  | 44.74  | 72.63               | 61.17  | 46.29  |
> |                  | WATT [3] | 89.28            | 60.22  | 46.50  | 78.36             | 79.40  | 45.02  | 72.69               | 60.87  | 46.49  |
> |                  | OSTTA [1] | 92.77           | 24.41  | 54.31  | 88.69             | 51.03  | 47.23  | 91.90               | 29.65  | 54.47  |
> |                  | ROSITA [5]  | 94.39             | 23.97  | 57.18  | 90.28             | 45.25  | 51.76  | 91.27               | 30.19  | 56.12  |
> |                  | Open‑IRT | **99.38**         | **3.71** | **58.75** | **94.09**     | **27.21** | **52.28** | **93.29**           | **22.30** | **56.97** |
>
> We observe that OSTTA shows the most notable improvement due to its alignment with open-set conditions. Tent and WATT primarily focus on test-time adaptation without explicitly modeling the distinction between ID and OOD data. Although OSTTA is more suitable for open-set scenarios, it still falls short in adaptability compared to more recent methods like [5] and our approach with 7.18 average FPR improvements.
>
> We acknowledge the important role these baselines have played in advancing the field, and we believe our work builds upon and extends their foundational contributions in the context of open-set, single-image TTA.
>
> ---
>
> ### **\[Weakness 3] Does Corruption Dataset Influence Experimental Results?**
>
> Thank you for your valuable suggestion. We carefully addressed this concern in our experimental design. In **Table 3 (Line 308)**, we present experiments with **uncorrupted** datasets, using **VisDA** as the ID dataset and **MNIST** and **SVHN** as OOD datasets—none of which are corrupted. Additionally, we evaluated settings where both ID and OOD data are corrupted, as shown in **Table 1 (Line 275)** and **Table 7 (Line 750)**, for **CIFAR-10C** and **CIFAR-100C**. For example, in the MapLe-based CIFAR-10C to CIFAR-100C experiment, our method outperforms the second-best method by 1.72 in FPR and improves accuracy by 0.41.
>
> Additionally, in response to **[Weakness 2]**, we conducted experiments with clean **CIFAR-10** and **CIFAR-100** datasets for ID, as well as **MNIST**, **SVHN**, and **Tiny-ImageNet** for OOD. Compared to the full ImageNet, we used Tiny-ImageNet for more efficient validation. These results further show that performance trends between our method and baselines remain consistent, regardless of corruption.
>
> In these settings, our method consistently performs well, demonstrating that its effectiveness **does not rely on data corruption**. Furthermore, if corruption introduces domain shifts, all methods would be equally affected, ensuring fairness in the evaluation.
>
> From a theoretical perspective, as outlined in **Hypothesis 2**, our method models the distance between ID, OOD, and the intermediate domain, driven by semantic-level distributional differences, which corruption does not fundamentally alter. For example, even if a cat image is corrupted, its core semantic representation as a "cat" remains intact.
>
> ---
>
> ### **\[Question 1] On the Choice of Harmonic Accuracy and Related Metrics**
>
> Thank you for your insightful comment. We agree that **harmonic accuracy alone does not explicitly separate ID classification from ID-OOD discrimination performance**, as it reflects only their overall balance.
>
> To address this, we include two additional metrics—**FPR\@95** (**Line 641**) and **AUROC** (**Line 627**)—to directly evaluate ID-OOD discrimination. Lower FPR\@95 and higher AUROC values indicate better separation between ID and OOD samples.
>
> Together, FPR\@95 and AUROC capture OOD detection performance, while harmonic accuracy helps reflect ID classification indirectly. This combination provides a more comprehensive assessment of model performance under open-world settings.
>
> ---
>
> ### **\[Question 2] Potential Impact of Corruption Dataset on Experimental Results**
>
> As discussed in **\[Weakness 3]** and supported by the experiments in **\[Weakness 2]**, we conclude that our method does not directly benefit from dataset corruption. The observed performance improvements stem from the effective modeling of the distance between ID and OOD distributions, as described in **Hypothesis 1 and Hypothesis 2**.
>
> ---
>
> ### **\[Question 3] Final Score Calculation via Equation 3**
>
> Thank you for your comment. Indeed, the final ID/OOD score is computed using **Equation 3**. To clarify, **Section 3** is structured as follows: **Section 3.1** focuses on the calculation of the ID/OOD score based on **Hypothesis 1**, while **Section 3.2** introduces the concept of the intermediate domain and the test-time adaptation method, which is developed around **Hypothesis 2**.
>
> ---
>
> ### **\[Question 4] Additional Experiments (Tent \[2], WATT \[3], OSTTA \[1])**
>
> Thank you for your suggestion regarding the experiments. As per your recommendation, we have included the requested experiments with **Tent \[2]**, **WATT \[3]**, and **OSTTA \[1]** in **\[Weakness 2]**. All experiments were conducted using **uncorrupted datasets**.
>
> ---
>
> ### **Reference**
>
> [1] Lee, Jungsoo, et al. "Towards open-set test-time adaptation utilizing the wisdom of crowds in entropy minimization." ICCV . 2023.
>
> [2] Wang, Dequan, et al. "Tent: Fully test-time adaptation by entropy minimization." arXiv preprint arXiv:2006.10726 (2020).
>
> [3] Osowiechi, David, et al. "WATT: Weight average test time adaptation of CLIP." Advances in neural information processing systems 37 (2024): 48015-48044.
>
> [4] Shu, Manli, et al. "Test-time prompt tuning for zero-shot generalization in vision-language models." NeurIPS 35 (2022): 14274-14289.
>
> [5] Sreenivas, Manogna, and Soma Biswas. "Efficient Open Set Single Image Test Time Adaptation of Vision Language Models." arXiv preprint arXiv:2406.00481 (2024).

---

> > ### Comment · Reviewer_g3am · 2025-08-01
> > **Follow-up discussion**
> >
> > Thank you for addressing my questions:
> >
> > __Q1:__ I understand that AUROC and FPR are the main OOD detection metrics, but also showing the ID accuracy is a standard: are these results available? The importance of this is that method purely dedicated to ID TTA may be very bad on OOD detection, but can still achieve a good ID accuracy, because this latter metric is often measured solely on the ID set. Having the standard accuracy allows to see how much a method can degenerate a model due to the presence of OOD samples.

---

> ### Author Response · Authors · 2025-08-02
> **Results of standard ID Accuarcy**
>
> **Thank you very much for your valuable suggestion and insightful feedback**. We do have the standard ID accuracy results, as they are used to compute the harmonic accuracy in our paper. We apologize for not including them during the rebuttal phase.
>
> Below, we provide the complete accuracy results, including the standard ID Accuracy, which clearly reflects whether a method experiences degradation on ID samples under the presence of OOD inputs during test-time. We hope this resolves your concern:
>
> | Method (ViT‑B32) |         | MNIST    | SVHN     | Tiny‑ImageNet |
> |:----------------:|:-------:|:--------:|:--------:|:-------------:|
> |                  |         | ID Acc   | ID Acc   | ID Acc        |
> | CIFAR‑10         | ZS‑Eval | 58.02    | 57.75    | 68.99         |
> |                  | Tent [2]| 62.09    | 63.79    | 71.79         |
> |                  | TPT [4] | 61.21    | 64.54    | 71.58         |
> |                  | OSTTA [1]| 67.70   | 65.83    | 71.69         |
> |                  | WATT [3]| 64.80    | 66.73    | 72.46         |
> |                  | ROSITA [5]| 70.27  | 64.13    | 72.68         |
> |                  | Open‑IRT| **70.87**| **67.83**| **72.89**     |
> | CIFAR‑100        | ZS‑Eval | 30.05    | 29.76    | 37.36         |
> |                  | Tent [2]| 30.10    | 29.69    | 37.48         |
> |                  | TPT [4] | 30.41    | 29.76    | 37.43         |
> |                  | OSTTA [1]| 36.98   | 36.02    | 41.37         |
> |                  | WATT [3]| 33.87    | 35.19    | 39.64         |
> |                  | ROSITA [5]| 41.87  | 38.09    | 45.88         |
> |                  | Open‑IRT| **42.00**| **40.11**| **45.98**     |
>
> The first table is derived from the results presented in our rebuttal **[Weakness 2]**, while the latter two are reproduced from **Table 3** and **Table 7** in our main paper. These results illustrate the performance of various methods on the ID dataset in the presence of OOD samples.
>
> From the tables, it is evident that despite the presence of OOD inputs, our method consistently achieves either the best or second-best performance in ID accuracy across multiple tasks. This further supports the effectiveness of our approach in maintaining ID performance while ensuring robust OOD detection.
>
> | Method | CIFAR-10C →CIFAR-100C    | CIFAR-100C→CIFAR-10C     |
> |:-------:|:--------:|:-------------:|
> |         | ID Acc   | ID Acc   |
> | ZS‑Eval |  60.88   |  32.06   |
> | TPT [4] |  61.25   |  32.00   |
> | OWTTT [6]| 59.36  |  30.29   |
> | TDA [7]| 65.79  |  37.20   |
> | UniEnt [8]| 64.32  |  35.04   |
> | ROSITA [5]| **67.28**  |  37.07   |
> | Open‑IRT| 65.94| **40.79**|
>
> | Method | VisDA→MNIST    | VisDA→MNISTM     |
> |:-------:|:--------:|:-------------:|
> |         | ID Acc   | ID Acc   |
> | ZS‑Eval |  64.37   |  64.30   |
> | TPT [4] |  64.56   |  64.51   |
> | TPT-C [4] |  60.18   |  66.49   |
> | ROSITA [5]| 82.87  |  78.05   |
> | Open‑IRT| **83.29**|**79.29** |
>
>
>
> [1] Lee, Jungsoo, et al. "Towards open-set test-time adaptation utilizing the wisdom of crowds in entropy minimization." ICCV . 2023.
>
> [2] Wang, Dequan, et al. "Tent: Fully test-time adaptation by entropy minimization." arXiv preprint arXiv:2006.10726 (2020).
>
> [3] Osowiechi, David, et al. "WATT: Weight average test time adaptation of CLIP." Advances in neural information processing systems 37 (2024): 48015-48044.
>
> [4] Shu, Manli, et al. "Test-time prompt tuning for zero-shot generalization in vision-language models." NeurIPS 35 (2022): 14274-14289.
>
> [5] Noisy Test-Time Adaptation in Vision-Language Models, ICLR 2025
>
> [6] Li, Yushu, et al. "On the robustness of open-world test-time training: Self-training with dynamic prototype expansion." ICCV. 2023.
>
> [7] Karmanov, Adilbek, et al. "Efficient test-time adaptation of vision-language models."  CVPR. 2024.
>
> [8] Gao, Zhengqing, Xu-Yao Zhang, and Cheng-Lin Liu. "Unified entropy optimization for open-set test-time adaptation." CVPR. 2024.

---

> ### Author Response · Authors · 2025-08-05
>
> Dear Reviewer,
>
> We sincerely appreciate your thoughtful question regarding the ID accuracy under the presence of OOD inputs. As a follow-up, we’ve provided detailed results addressing this point, demonstrating that our method maintains strong ID accuracy while achieving robust OOD detection.
>
> Since the discussion phase is approaching its end, we would like to kindly check if there are any further questions or concerns we can help clarify.
>
> Thank you again for your valuable feedback and time.

---

> ### Comment · Reviewer_g3am · 2025-08-05
> **Follow up and closing**
>
> Thanks to the authors for their efforts in responding to my questions.
>
> In regard of the competitive results in ID accuracy, I believe their contribution is valuable. I think that harmonic accuracy could become an interesting standard metric for the future of OOD TTA, but encourage the authors to include these ID results in a next version of the paper.
>
> In light of the provided evidence and the authors' effort in responding my concerns and those from other reviewers, and as a contribution to OOD TTA (a field that is just starting to be more widely explored in VLMs), I would increase my score to borderline accept.

---

> > ### Author Response · Authors · 2025-08-05
> > **Response and Appreciation to Reviewer g3am**
> >
> > Dear Reviewer g3am,
> >
> > **We truly appreciate your recognition of our contributions in OOD TTA**, especially for the significance of the proposed harmonic accuracy metric. We are also deeply grateful for your increased score and your encouraging words.
> >
> > **In response to your suggestion**, we fully agree that including ID accuracy results would further enhance the completeness and clarity of our work. In the revised version, we will explicitly report ID accuracy in the main paper. We have reviewed the space constraints and confirm that there is sufficient room for this addition. Furthermore, we will provide more comprehensive results in the supplementary material to better support our claims and ensure reproducibility.
> >
> > **Thank you again for your valuable time and insights.** Your support and recognition serve as great encouragement for us. **We wish you continued success in your research and all the best in life.**
> >
> > Best regards,
> >
> > All authors

---

### Official Review · Reviewer_4p14 · 2025-07-01

**Clarity:** 4
**Significance:** 3
**Originality:** 2
**Rating:** 4
**Confidence:** 4

**Summary:**

The authors propose a novel open-set test-time adaptation method designed to handle single-image inputs. The approach first classifies each test sample as in-distribution (ID) or out-of-distribution (OOD) using the Polarity-aware Prompt-based OOD Filtering (PPF) module, which computes similarity scores with both positive and negative prompts. These scores are modeled with a Gaussian Mixture Model (GMM) to distinguish ID from OOD samples. Once classified, the model performs adaptation via an Intermediate Domain-based Test-time adaptation (IDT) module, which constructs a middle domain between ID and OOD representations. Adaptation is guided by a contrastive loss and a bidirectional pseudo-labeling strategy to refine the feature space and improve ID-OOD separation over time.

**Questions:**

1. The supplementary material discusses performance across different batch sizes, but does not report results for very small batches (e.g., B=1 or B=8). Could the authors include results in that regime, or comment on how the method degrades with decreasing sample support?

2. The method relies on negative prompts (e.g., "a photo of no [CLS]") for OOD detection. However, prior work (e.g., CLIPN, NegPrompt) reports that such prompts can be unreliable with CLIP. Could the authors provide more insight into how their specific formulation avoids these pitfalls, or whether other prompt designs were tested?

3. Could the authors elaborate on the design and selection of prompts (positive and negative)? For example, how sensitive is the method to the phrasing of prompts? Did they explore alternatives such as learned or templated prompt tuning?

4. During early test-time steps when only a few samples are available, how stable is the GMM estimation? Could the authors elaborate on how bootstrapped resampling helps here, and whether other techniques were considered?

**Ethical Concerns:**

["NO or VERY MINOR ethics concerns only"]

**Final Justification:**

Thank you for the detailed and thoughtful rebuttal. I appreciate the additional experiments and clarifications provided, which help to address some of the concerns I raised.

That said, I remain somewhat concerned about the method’s reliance on a sufficiently large memory bank for GMM estimation. While the reported results suggest that performance stabilizes for B≥16, this requirement appears to contradict the claim that the method is applicable in a truly single-image test-time scenario. In practice, needing several images beforehand seems to weaken the core promise of single-image adaptability. I would encourage the authors to clarify this nuance more explicitly in the paper to avoid potential overstatement.

**Limitations:**

yes

**Paper Formatting Concerns:**

No Formatting Concerns

**Quality:**

3

**Strengths And Weaknesses:**

Strengths:
- The paper is well written and clearly structured, making the proposed method easy to understand and potentially straightforward to reproduce.
- The approach is extensively compared against a wide range of state-of-the-art baselines across multiple datasets, consistently showing strong performance.
- The authors conduct several relevant ablation studies, including the impact of different score modeling methods (e.g., GMM vs. LDA) and the adaptation loss components.
- The supplementary material is thorough and provides additional insights, which strengthens the overall experimental validation.

Weaknesses:
- Although the method targets the single-image test-time adaptation setting, it still relies on feature and score banks that accumulate information over multiple samples. Therefore, the title and claims of true single-image operation may be slightly overstated. In addition, the supplementary study on batch size does not explore values below B=64; evaluating lower values would be important to fully support the single-image claim, even if performance degrades.
- The method employs negative prompts such as “a photo of no [CLS]” to distinguish OOD samples. However, recent works (e.g., Learning Transferable Negative Prompts for OOD Detection, 2024; CLIPN for Zero-Shot OOD Detection, 2023) suggest that CLIP often struggles with such formulations. A more detailed discussion on prompt engineering choices and their limitations would strengthen the paper and help clarify whether this design choice generalizes well.
- While the authors introduce bootstrapped resampling to stabilize GMM estimation with few samples, there is limited quantitative or qualitative evidence that this improves performance. It would be useful to understand how sensitive the method is to GMM misestimation early in adaptation.

---

> ### Author Rebuttal · Authors · 2025-07-28
>
> ### **Thank you for the valuable time you spent reviewing our manuscript. We appreciate your recognition of the readability of our writing and the thoroughness of our experiments.**
>
> Thank you for your insightful comments. In response, we have conducted additional verification experiments. We believe your concerns can be addressed through these experiments and the additional clarifications provided. **Key points** have been highlighted in **bold** to help you easily locate the relevant parts in the paper.
>
>
> ---
>
> ### **\[Weakness 1] Performance with Small B Values (B<64)**
>
> We appreciate your careful reading, including the supplementary materials. Your point is valid, and we have addressed it by including experiments with different values of **B = 4, 8, 16, 32, 64, 128, 256, 512, 1024** based on CLIP, specifically for the CIFAR-10C to MNIST task. The results are presented in the table below:
>
> ---
> | B        | 4      | 8      | 16     | 32     | 64     | 128    | 256    | 512    | 1024  |
> | :------- | :----- | :----- | :----- | :----- | :----- | :----- | :----- | :----- | :----- |
> | Accuracy  | 81\.35 | 81\.65 | 83\.83 | 84\.26 | 84\.40 | 84\.53 | 84\.55 | 84\.67 | 84\.48 |
>
> The results confirm that extremely small values of B lead to performance degradation due to insufficient data for reliable GMM estimation. However, when B ≥ 16, the accuracy stabilizes, indicating that our method is robust under a wide range of practical settings.
>
>
> ---
> ### **\[Weakness 2] Effectiveness of using negative prompts**
> Your question is critical, and we have carefully considered this aspect in both the method design and theoretical analysis. As detailed in **Section A.2** of the supplementary materials, we address the theoretical validity of the negative prompt. Specifically, in **Line 525**, we acknowledge that VLMs face known challenges in implementing human-like negation understanding [3]. However, the effectiveness of the negative polarity prompt in our framework does not entirely depend on the perfect negative logical reasoning ability of the VLM. Instead, it leverages the differential representation capabilities of the text encoder: the vocabulary difference between the positive and negative prompts, $w_c^+$ and $w_c^-$, naturally leads to different embedding vectors $p_c$ and $p_c'$ in the shared embedding space. Our framework utilizes this induced representational difference as a discriminative signal for OOD detection. This idea is reflected in the design of the polarity gap item in **Equation 3** (**Line 158**). It is also intuitively visualized in **Figure 3**.
>
>
> ---
>
> ### **\[Weakness 3] Effectiveness of GMM in the Early Adaptation Phase**
>
> Thank you for raising this insightful point. From a technical standpoint, during the early adaptation stage of our method, we deliberately delay the fitting of the GMM and the computation of the corresponding loss until a sufficient number of samples (i.e., bank size $B$) has been accumulated. In other words, to avoid the uncertainty introduced by small sample sizes propagating to the model through loss gradients, we perform direct inference for those samples instead, without engaging in backpropagation. We acknowledge that predictions for these samples may be relatively weak.
>
> Therefore, there are two main motivations behind incorporating the bootstrapped resampling method:
>
> 1. OOD detection inherently depends on modeling data distributions, which requires a sufficient number of samples to distinguish ID from OOD instances effectively.
>
> 2. GMM fitting relies on the statistical properties of the data; with limited samples, parameter estimates may deviate from the true distribution, degrading performance.
>
> The following table presents the experimental results to support this design:
>
> | **B**    | **4**   | **8**   | **16**   | **32**  | **64**  | **128**  | **256** | **512** | **1024** |
> |----------|---------|---------|---------|---------|---------|---------|---------|---------|---------|
> | **w/ Resample**     | 81.35  | 81.65  | 83.83  | 84.26  | 84.40  | 84.53  | 84.55  | 84.67  | 84.48  |
> | **w/o Resample**    | 81.24  | 81.54  | 83.78  | 84.24  | 84.38  | 84.52  | 84.54  | 84.67  | 84.48  |
> | **Δ Accuracy (↑)**       | +0.11  | +0.11  | +0.05  | +0.02  | +0.02  | +0.01  | +0.01  | +0.00  | +0.00  |
>
> Based on our experiments under the conditions of $B = 4$ and $B = 8$, we effectively simulate scenarios where the memory bank contains only a small number of samples during the early adaptation stage. The results demonstrate that the bootstrapped resampling strategy successfully mitigates performance degradation caused by insufficient data during GMM fitting.
>
> Moreover, when the bank size $B$ is sufficiently large, the impact of bootstrapped resampling becomes marginal. This is expected, as no loss backpropagation occurs during the early inference stage, and thus, performance is less sensitive to sample size once adequate data is accumulated.
>
> Overall, our experiments suggest that as long as a reasonable $B$ value is chosen—avoiding extremely small settings—the proposed method remains robust and performs consistently across different configurations.
>
>
> ---
>
> ### **\[Question 1] Performance under Very Small B**
>
> As discussed in **\[Weakness 1]**, model performance degrades when $B$ is very small, primarily because GMM fitting requires a minimum number of samples to produce reliable estimates. However, we observe that when $B > 16$, the model performance stabilizes and reaches saturation.
>
> ---
>
> ### **\[Question 2] Theoretical Validity of Negative Prompts**
>
> As discussed in **\[Weakness 2]**, we elaborate on the theoretical rationale of using negative prompts in **Section A.2** of the supplementary material (**Line 525**). Our method does not rely on the VLM’s precise understanding of semantic negation. Instead, it leverages the **representational difference** between positive and negative prompts.
>
> ---
>
> ### **\[Question 3] Choice of Prompt Format**
>
> Yes, we explored the prompt format based on the discussion in CLIPN \[1]. In practice, vision-language models (VLMs) are known to be sensitive to the phrasing of input prompts, and carefully crafting prompt formats can indeed improve performance. However, we intentionally adopted a simple and consistent format — “*a photo of no \<class\>*” — for two key reasons.
>
> 1. We aimed to ensure fairness across all baseline comparisons. All baselines use the prompt “*a photo of a \<class\>*” for inference. To maintain consistency and minimize bias, we chose a minimal modification of that format for negative prompts.
>
> 2. All methods, including ours and baselines, utilize the same CLIP-based text encoder. Since a well-optimized prompt would benefit all methods equally, we argue that the main performance differences stem from algorithmic and theoretical innovations, rather than from prompt engineering.
>
> Our method focuses on data-level improvements by constructing an intermediate domain. This perspective is theoretically and practically complementary to works based on learnable or tunable prompts \[4–6]. As such, our contributions emphasize data-driven design and theoretical insights. We also briefly discuss the learnable module option in the supplementary material (**Line 665**).
>
> We now provide results on CIFAR-100C to MNIST using negative prompts introduced in CLIPN \[1], as follows:
>
> ---
>
> | Prompt Phrase                     | Accuracy (%) |
> |----------------------------------|--------------|
> | a photo of no `<class>`          | 62.01        |
> | a photo without `<class>`        | 61.97        |
> | a photo not appearing `<class>`  | 62.12        |
> | a photo not contain `<class>`    | 62.19        |
>
> From the experimental results, we observe that different prompt formats do slightly impact model performance. This is due to the sensitivity of **Equation 3** to the input text prompts, where different textual inputs lead to different computed scores. Notably, some prompts even yield better performance than our original formulation.
>
> Although different negative prompts lead to marginal performance differences (±0.1%), we intentionally avoided prompt optimization to ensure fair comparisons. This further highlights that our gains stem from algorithmic contributions rather than prompt engineering.
>
> ---
>
> ### **\[Question 4] Effectiveness of GMM in the Early Stage**
>
> As discussed in **\[Weakness 3]**, we employ bootstrapped resampling to address the challenge of limited samples during the early stages of training. This design is motivated by two key considerations: (1) OOD detection fundamentally relies on characterizing the distributional differences between ID and OOD data, which becomes more effective as the sample size increases; and (2) GMM fitting depends on sufficient data to reliably estimate the underlying statistical properties of the distribution.
>
> As shown in the experiments under **\[Weakness 3]**, once a sufficient number of samples is accumulated, the effect of this strategy becomes negligible in terms of performance improvement. From a theoretical perspective, alternative sampling strategies could also serve to increase the diversity of the feature bank and alleviate the sparse-sample issue in early training stages.
>
> ---
>
> ### **Reference**
> [1] Wang, Hualiang, et al. "Clipn for zero-shot ood detection: Teaching clip to say no." ICCV. 2023.
>
> [2] Yang, Jingkang, et al. "Generalized out-of-distribution detection: A survey." IJCV 132.12 (2024): 5635-5662.
>
> [3] Li, Tianqi, et al. "Learning transferable negative prompts for out-of-distribution detection." CVPR. 2024.
>
> [4] Zhou, Kaiyang, et al. "Learning to prompt for vision-language models." IJCV 130.9 (2022): 2337-2348.
>
> [5] Zhou, Kaiyang, et al. "Conditional prompt learning for vision-language models." CVPR . 2022.
>
> [6] Kan, Baoshuo, et al. "Knowledge-aware prompt tuning for generalizable vision-language models." ICCV. 2023.

---

> ### Comment · Reviewer_4p14 · 2025-08-04
>
> Thank you for the detailed and thoughtful rebuttal. I appreciate the additional experiments and clarifications provided, which help to address some of the concerns I raised.
>
> That said, I remain somewhat concerned about the method’s reliance on a sufficiently large memory bank for GMM estimation. While the reported results suggest that performance stabilizes for B≥16, this requirement appears to contradict the claim that the method is applicable in a truly single-image test-time scenario. In practice, needing several images beforehand seems to weaken the core promise of single-image adaptability. I would encourage the authors to clarify this nuance more explicitly in the paper to avoid potential overstatement.

---

> > ### Author Response · Authors · 2025-08-04
> >
> > **We truly appreciate the careful reading of our rebuttal and the thoughtful feedback, which have significantly helped us further reflect on and improve the clarity of our work.**
> >
> > We will revise the paper accordingly to present a more accurate and balanced description. In our work, the core contribution lies in the theoretical proposal of the intermediate domain hypothesis (**Hypothesis 2**) and the separation principle between ID and OOD data (**Hypothesis 1**). From a methodological perspective, these theoretical insights are applicable in **both single-image and multi-image scenarios**.
> >
> > To avoid the practical contradiction between requiring multiple images and the intended single-image test-time setting, we employ an updatable feature memory bank to accumulate features from incoming test samples (**Line 192**). These stored features are not involved in the gradient backpropagation. This approach avoids directly storing image batches and aligns with the widely adopted practice of maintaining a feature memory bank for test-time adaptation [1–3].
> >
> > That said, we acknowledge that since our paradigm depends on modeling the overall data distribution, gradient-based updates cannot be performed until a minimum number of features are accumulated in the memory bank. During this initial phase, the model only performs inference. Although only 16 features are needed to trigger updates—which is minimal compared to the total number of samples—we agree that this nuance is important and deserves clarification. We are also actively exploring approaches that treat all samples equally from the very beginning.
> >
> > We thank the reviewer once again for the valuable suggestions, and we will make sure to address this point explicitly and carefully in the revised version of the paper.
> >
> > ### **References**
> >
> > [1] Wang, Shuai, et al. "Feature alignment and uniformity for test time adaptation." CVPR, 2023.
> >
> > [2] Yuan, Longhui, Binhui Xie, and Shuang Li. "Robust test-time adaptation in dynamic scenarios." CVPR, 2023.
> >
> > [3] Chen, Dian, et al. "Contrastive test-time adaptation." CVPR, 2022.

---

### Official Review · Reviewer_YfPH · 2025-07-02

**Clarity:** 1
**Significance:** 2
**Originality:** 2
**Rating:** 4
**Confidence:** 4

**Summary:**

This paper addresses the problem of open-set single-image test-time adaptation (TTA), where in-distribution (ID) and out-of-distribution (OOD) samples appear unpredictably. The proposed framework comprises two main components: (1) Polarity-aware Prompt-based OOD Filter (PPF) and (2) Intermediate Domain-based Test-time adaptation (IDT). The PPF computes a sample-wise score by combining absolute semantic alignment (between image features and positive prompts) and relative semantic polarity (between image features and corresponding positive/negative prompts). This score is then modeled via a Gaussian Mixture Model (GMM) to classify samples as either ID or OOD. The IDT constructs intermediate domain features and uses contrastive learning (via InfoNCE loss) to encourage separation between ID and OOD features from the intermediate representations.

**Questions:**

Refer to weaknesses above.

Additional question: How are the positive and negative prompts learned?

**Ethical Concerns:**

["NO or VERY MINOR ethics concerns only"]

**Final Justification:**

Overall, the rebuttal has significantly improved my understanding of the paper and strengthened the value of the proposed approach. Consequently, I will revise my recommendation to borderline acceptance.

**Limitations:**

N.A.

**Paper Formatting Concerns:**

N.A.

**Quality:**

2

**Strengths And Weaknesses:**

* Strengths
1. The paper introduces a novel and practically relevant setting: open-set single-image test-time adaptation.
2. The proposed method performs competitively on standard benchmarks.

* Weaknesses
1. Technical clarity and presentation

1-1. The introduction lacks intuitive motivation. For example, the rationale behind using intermediate domain features to distinguish ID and OOD samples is not clearly justified. The term “polarity” is also introduced abruptly and is only clarified later in (3).

1-2. Lines 144–148 just provide a description of the method, but the underlying motivation and the importance of the proposed design are not sufficiently justified.

1-3. There are mismatches between terminology used in figures and text. For instance, the terms 'Sim_pos’, 'Sim_neg’, and 'Sim_diff’ in Figure 2 are not consistently reflected in the main text. `Prototypes’ in Figure 2 are mentioned as prompts in the main text.

1-4. The sliding window strategy is not clearly defined. Is it equivalent to a FIFO queue? This should be clarified.

1-5. $f_s$ is a multi-dimensional vector, so applying variance directly is questionable; covariance may be more appropriate unless the computation is done per-channel. This needs clarification.

1-6. While (8) uses samples that satisfy ($y^+ = y_p$), this constraint is not applied to (9). The motivation behind this inconsistency should be discussed.

1-7. What happens when the sample does not belong to the two cases in (7), (10), and (13), i.e., $\mu_O <= Conf(x) <= \mu_I$? The current decision rule does not account for such cases.


2. In (13), the use of $-L_{CE}$ for OOD samples aims to suppress intra-class noise, but this may unintentionally distort OOD representations.

3. The method depends heavily on the threshold parameters $\mu_I$ and $\mu_O$. Empirical analysis of sensitivity to these hyperparameters should be provided.

4. L202: While the proposed method attempts to distance both ID and OOD samples from the intermediate domain representations, it is possible that ID and OOD samples may still be drawn closer to each other (e.g., if vectors $F_I - F_M$ and $F_O - F_M$ are aligned). This could undermine the objective of separation and should be theoretically or empirically addressed.

* Overall evaluation

Overall, this paper tackles a challenging and underexplored problem in open-set single-image TTA and provides an reasonable approach leveraging vision-language models. However, the current version suffers from unclear technical presentation and insufficient explanation of key design choices.

---

> ### Author Rebuttal · Authors · 2025-07-27
>
> ### **Thank you for your time and consideration. We appreciate your recognition of the novelty and practicality of our method.**
>
> We believe that your concerns can be addressed by providing further clarification of the technical details, allowing us to reach a mutual understanding.  We would like to highlight the **original details** corresponding to the **bolded lines** in the rebuttal section, as they provide further clarification and location in paper. Your attention to these details would be greatly appreciated.
>
> ---
>
> ### **\[1-1] Motivation Behind the Method in the Introduction**
>
> The motivation behind our Open-IRT method is intuitively presented in **Figure 1** and its caption (**Line 47**) and **Hypothesis 1 & 2**: the separation of ID and OOD distances through an intermediate domain, with the focus on expanding the distances within the separated regions to achieve a better distinction. Due to space limitations in the Introduction, we were unable to elaborate on the complete motivation for the method or specific terms (such as "polarity"). Therefore, the full motivation for our method is elaborated in the **Methodology Section**, particularly around **Hypothesis 1 (Line 47)** and **Hypothesis 2 (Line 191)**.
>
> For **Hypothesis 1**, we design an ID/OOD scoring system based on the differential representation capabilities of the visual-language model's text encoder for prompts. In **Hypothesis 2**, we construct an intermediate domain. To substantiate the validity of the intermediate domain, we provide both **Theoretical Justification (Line 559)** and experimental verification (**Line 323**) in the appendix, along with a clear visualization of the intermediate domain (**Figure 4**).
>
> ---
>
> ### **\[1-2] L144-L148 Motivation of the Method in the Introduction Section**
>
> As discussed in **[1-1]**, we provided only a brief overview of the motivation for the method in **Figure 1 (Line 47)**. A more detailed explanation is found in **Hypothesis 1 (Line 47)**, and **Hypothesis 2 (Line 191)**. The theoretical foundations for the method are further analyzed and validated in the supplementary materials (**Lines 487-583**).
>
> ---
>
> ### **\[1-3] Explanation of Symbols in Figure 1**
> Due to the need for clarity and conciseness, we opted for abbreviations in **Figure 1**. This process corresponds to **Equation 3 (Line 164)**, where:
>
> $sim\_{pos} = sim(f, p\_c)$, $sim\_{neg} = sim(f, p_c')$, $sim\_{diff} = \left| sim(f, p\_c) - sim(f, p\_c') \right|$
>
> These quantities are used to compute scores in Equation 3. In the context of the paper, **Prototypes** refers to the embedded prompt tensors.
>
> ---
>
> ### **\[1-4] Definition of Sliding Window Strategy**
>
> We adopt the standard definition of Sliding Window in sequence data processing from computer science \[1], where, in the context of our paper, it functions equivalently to a FIFO queue.
>
> ---
>
> ### **\[1-5] $f\_s$ Computation Paradigm**
>
> The method for calculating the variance of $f_s$ has been validated for its rationality in \[2], where it is computed across spatial dimensions independently for each feature channel. The feasibility of this approach lies in preserving channel-specific style information. The variance for each channel emphasizes fine-grained structural characteristics in that feature dimension, such as color, texture intensity, etc., without mixing with other channels, thereby achieving refined style representation and alignment.
>
> ---
>
> ### **\[1-6] Differences between Equation 8 and Equation 9**
>
> The differences between **Equation 8** and **Equation 9** arise from the distinction between **ID** and **OOD** categories in the standard **OOD detection** task \[3]. The overall setup of the OOD detection task is to first perform **ID/OOD** binary classification, followed by internal classification within **ID** data \[3], without considering internal classification for **OOD** data. Therefore, in **Equation 8**, for ID data, the model needs to learn fine-grained semantic consistency and achieve precision for each class, which is why we set $y^{+} = \hat{y}\_{p}$, where $\hat{y}\_{p}$ is the pseudo-label designed for ID data (**Line 239**). Similarly, in the experiments, our accuracy metrics are also based on class-level accuracies (**Line 262, 655**). For OOD samples, since their class labels differ from those of ID samples, including $y^{+} = \hat{y}\_{p}$ in the calculation would render this condition meaningless.
>
> We assume that the distinction between fine-grained classification for ID data and binary classification for OOD data is well-understood in the context of OOD Detection. Therefore, we did not elaborate further in the main text.
>
> ---
>
> ### **\[1-7] Exceptions for $\text{Conf}(x) > \mu_{I}$ or $\text{Conf}(x) < \mu_{O}$**
> As shown directly in **Equation 13**, the loss term $L\_{psd}$ is calculated only when $\text{Conf}(x) > \mu_{I}$ or $\text{Conf}(x) < \mu_{O}$; otherwise, there is no backpropagation of loss, and the sample is directly used for inference (**Line 251**). The same applies to **Equations 7 and 10**. This approach avoids introducing noisy information due to ambiguous confidence into the model's learning process. For clarity, we will explicitly state that the loss is zero when the conditions $\text{Conf}(x) \leq \mu_I$ and $\text{Conf}(x) \geq \mu_O$ are met.
>
> ---
>
> ### **\[2] The Impact of $-L\_{CE}$ on OOD Samples**
>
> In fact, for a standard **OOD Detection** task setup \[3], we only need to perform binary classification between **ID** and **OOD**, and focus on **class-level** classification for **ID** samples without needing to perform fine-grained classification for **OOD** samples. As shown in **Figure 6**, existing OOD Detection methods (TPT[4], ROSITA[5], Open-IRT) introduce some distortion in the feature distribution of OOD data compared to direct inference (ZS-Eval). This is a well-known phenomenon in this field and does not negatively impact the evaluation metrics.
>
> Additionally, it can be observed that our **$-L\_{CE}$** method pushes samples away from intra-class noise. Consequently, the resulting impact on the distribution of **OOD** samples is a normal occurrence and does not interfere with the **ID-OOD** binary classification, nor does it affect the fine-grained classification results within **ID** classes.
>
> ---
>
>
> ### **\[3] Values of $\mu_I$ and $\mu_O$, and Hyperparameters Robustness Experiment**
>
> In fact, $\mu_I$ and $\mu_O$ are not hyperparameters. Their definitions are provided in **Line 216**, where they are dynamically calculated mean values based on the confidence of the samples. By dynamically computing the confidence mean, we are able to adjust what is considered high confidence and low confidence according to different distributions, thus avoiding the limitations of fixed thresholds.
>
> Furthermore, the main performance improvement in our method is driven by the two theoretical mechanisms proposed in **Hypotheses 1 & 2**. As seen in the ablation study (**Line 317**), the choice of threshold is merely a technical means in the implementation process and is not the primary contribution.
>
> Additionally, other hyperparameter experiments (B, K, $\alpha$, $\lambda$) can be found in Sections C.2 (Line 770) and C.3 (Line 779) of the supplementary materials. Our method demonstrates robustness with respect to these hyperparameters. For example, the accuracy fluctuation due to $\alpha$ values ranging from 0.1 to 0.5 is less than 0.33%, and for $\lambda$ values between 0.1 and 0.5, the accuracy fluctuation is less than 0.57%.
>
> ---
>
> ### **\[4] Optimization Goal and Theory for the Distances between $\mathcal{F}_I$, $\mathcal{F}_M$, and $\mathcal{F}_O$**
>
> By examining the optimization goals in **Equation 8 & 9** (**Line 233 & 235**), it can be seen that they separately promote the separation between $\mathcal{F}_I$ and $\mathcal{F}_M$, as well as between $\mathcal{F}_O$ and $\mathcal{F}_M$. They also facilitate the separation of $\mathcal{F}_I$ and $\mathcal{F}_O$ through contrastive learning of positive and negative sample pairs. This fundamentally prevents the alignment between $\mathcal{F}_I - \mathcal{F}_M$ and $\mathcal{F}_O - \mathcal{F}_M$. Our method consistently improves FPR metrics, demonstrating better ID and OOD separation (**Figure 6**). As shown in Table 1 (CIFAR-10C to SVHN with CLIP), it boosts FPR95 by 12.88% over the second-best method.
>
>
> Based on **Line 208**, we visualize the result of the intermediate domain in **Figure 4**. It is clear from the figure that, according to our construction method, the intermediate domain is theoretically situated precisely between **ID** and **OOD**, which aligns with the situation set in the theoretical analysis of the supplementary materials (**Line 559**). This intuitively demonstrates the theoretical foundation of Hypothesis 2.
>
> ---
>
> ### **\[Additional question] How is the prompt learned?**
>
> The definitions of prompts are given in **Line 152**. We did not design updatable prompts because the core contribution of our paper lies in constructing an intermediate domain from the data perspective, and this design does not conflict with the use of updatable prompts. A detailed discussion of the update module can be found in **Line 665** of the supplementary materials.
>
> ---
>
> ### **Reference**
> [1] Ganardi, Moses, Danny Hucke, and Markus Lohrey. "Derandomization for Sliding Window Algorithms with Strict Correctness∗." Theory of Computing Systems 65.3 (2021): 1-18.
>
> [2] Huang, Xun, and Serge Belongie. "Arbitrary style transfer in real-time with adaptive instance normalization." ICCV. 2017.
>
> [3] Yang, Jingkang, et al. "Generalized out-of-distribution detection: A survey." IJCV 132.12 (2024): 5635-5662.
>
> [4] Shu, Manli, et al. "Test-time prompt tuning for zero-shot generalization in vision-language models." NeurIPS 35 (2022): 14274-14289.
>
> [5] Sreenivas, Manogna, and Soma Biswas. "Efficient Open Set Single Image Test Time Adaptation of Vision Language Models." arXiv preprint arXiv:2406.00481 (2024).

---

> > ### Comment · Reviewer_YfPH · 2025-08-08
> > **Response to rebuttal**
> >
> > Thank you for your detailed rebuttal.
> >
> > * **W1 (from 1-1 to 1-7)**
> >
> > I now have a clear understanding of the key motivation and design choices. I appreciate the detailed explanation, and I recommend revising the manuscript to incorporate the clarifications provided in the rebuttal.
> >
> > * **W4**
> >
> > This point has also been satisfactorily clarified.
> >
> >
> > Overall, the rebuttal has significantly improved my understanding of the paper and strengthened the value of the proposed approach. Consequently, I will revise my recommendation to borderline acceptance.

---

> ### Author Response · Authors · 2025-08-05
> **Further Clarification**
>
> **First, we sincerely thank you for recognizing the novelty of our method and the thoroughness of our experiments.** We appreciate your detailed comments and understand the concerns regarding clarity and design motivation. In addition to our previous response, we would like to briefly reiterate the key clarifications and confirm that all identified issues can be resolved in the revised version:
>
> * The rationale for using an intermediate domain is to create a separating margin that increases the distance between ID and OOD features, as supported by both theoretical analysis (**Appx. Line 559**) and visualizations (**Figure 4**).
> * The "polarity" score combines absolute and relative semantic alignments (**Equation 3**), computed via similarity between image and prompt embeddings (Sim_pos, Sim_neg, Sim_diff).
> * The sliding window is implemented as a standard FIFO queue. Variance is computed per channel to retain fine-grained feature structure.
> * Thresholds are not hyperparameters, but dynamically derived from sample confidence (**Line 216**).
> * Losses are applied only when specific conditions (**Equation 7/10/13**) are met; otherwise, no update is performed to avoid noisy gradients.
> * Prompts are predefined and not learned in this work, though our framework supports integration with learnable prompt modules (**Appx. Line 665**).
>
> We are committed to addressing these presentation issues—such as clearer motivation, terminology consistency, and explanation of loss conditions—in the revised manuscript. These improvements are straightforward and will significantly enhance clarity without altering the core method. Thank you again for your constructive feedback.

---

> ### Author Response · Authors · 2025-08-07
> **Follow-up on Reviewer Feedback and Clarifications for the Paper**
>
> Dear Reviewer,
>
> **We sincerely thank you again for your valuable comments on our paper and your recognition of the novelty of our method and the thoroughness of our experiments.** We truly appreciate the time and effort you have devoted to reviewing our work.
>
> We fully understand that you may have a very busy schedule, but we would be extremely grateful if you could kindly take a moment to review our responses to your previous comments. We have carefully addressed your concerns, especially those related to clarity, and have prepared corresponding revisions that we believe significantly improve the manuscript.
>
> Your feedback is very important to us. If you have any further suggestions or could confirm whether our clarifications resolve the issues you raised, it would be immensely helpful in guiding our revision.
>
> **Thank you again for your time and consideration. We greatly appreciate your support.**

---

> ### Author Response · Authors · 2025-08-08
> **Appreciation for Your Feedback**
>
> Dear Reviewer,
>
> **We sincerely thank you for your feedback throughout the review process.** We are truly grateful for the time and effort you devoted to carefully reading our manuscript and thoughtfully engaging with our rebuttal.
>
> Your comments have been instrumental in helping us identify and address areas for improvement. We are committed to fully incorporating the clarifications and enhancements discussed in the rebuttal—particularly in strengthening the introduction, refining terminology, and improving technical presentation—into the revised version.
>
> We deeply appreciate your updated evaluation and your willingness to reconsider your recommendation. **We wish you continued success in your research and all the best in your professional endeavors.**

---

### Official Review · Reviewer_c5VY · 2025-07-02

**Clarity:** 3
**Significance:** 3
**Originality:** 3
**Rating:** 4
**Confidence:** 4

**Summary:**

This paper addresses the challenge of open-set single-image test-time adaptation and introduces Open-IRT that incorporates a Polarity-aware Prompt-based OOD Filter (PPF) with an Intermediate Domain-based Test-time Adaptation (IDT) module. Instead of directly maximizing the distance between ID and OOD feature representations, this module constructs a "middle domain" in the feature space and optimizes a loss function that pushes both ID and OOD features away from this intermediate barrier. Experiments show that Open-IRT outperforms strong baselines on existing benchmarks.

**Questions:**

1. What are the performances compared with the most recent baseline [1]?
2. There are too many hyperparameters (bank size B, K, alpha, lambda, as well as mu_I and mu_O in eq13). How are they selected?
3. Detailed analysis of the computational latency per sample?

[1] Noisy Test-Time Adaptation in Vision-Language Models, ICLR 2025

**Ethical Concerns:**

["NO or VERY MINOR ethics concerns only"]

**Final Justification:**

Initially, I have concerns about performance improvement, complexity, and compared with the SOTA. The author addressed my concerns well. Therefore, I recommend borderline acceptance. The authors should include the additional experiments in the paper.

**Limitations:**

yes

**Paper Formatting Concerns:**

no major formatting issues

**Quality:**

3

**Strengths And Weaknesses:**

Strengths
1. The paper addresses the Open-Set Single-Image Test Time Adaptation problem with Vision Language Models, which is a challenging and practical scenario.
2. The paper is well written and easy to follow.
3. The paper provides extensive experiments, showing the effectiveness and versatility of the proposed method.
4. The idea of constructing middle domain is interesting.

Weaknesses
1. Lack of important comparision with most recent baseline [1].
2. The improvement is limited compare to the best baseline (Table 1 and 3), and is even worse than the best baseline in some cases.
3. The proposed Open-IRT framework is quite complex, involving multiple modules, feature/score banks, GMMs, KNN searches, and several loss terms.

[1] Noisy Test-Time Adaptation in Vision-Language Models, ICLR 2025

---

> ### Author Rebuttal · Authors · 2025-07-27
>
> ### **Thank you for your valuable time and effort in reviewing our manuscript. We appreciate your positive feedback on our idea, as well as your recognition of the clarity and thoroughness of our writing and experimental design.**
>
> We truly appreciate your thoughtful concerns. We believe that your concerns can be addressed through clarifications and further explanations in paper details. Below, we provide our clarification in detail, organized to address each of your concerns.
> We have highlighted the key points in **bold** to enhance your reading experience and make it easier to navigate through the **important points**.
>
> ---
>
>
> ### **\[Weakness 1] Additional Experimental Comparison of AdaND**
>
> Thank you for your suggestion. We have noted the work by \[1]. It introduces a new setting, **Zero-shot Noisy Test-time Adaptation (ZS-NTTA)** , which is different from ours. It adapt to noisy samples during test-time, focusing on noisy sample detection and classification accuracy.
>
> Our setting does not involve noisy conditions, so we did not run this experiment in the main body. However, we have incorporated this framework into our dataset and conducted additional experiments during the rebuttal phase. Despite improvements in the latest baseline, our method remains superior, such as 7.18 average FPR improvement, because our method excels in traditional single image OOD Detection tasks.
>
> ---
>
> | Method (ViT‑B32) |         | MNIST             |        |        | SVHN              |        |        | Tiny‑ImageNet       |        |        |
> |:----------------:|:-------:|:-----------------:|:------:|:------:|:-----------------:|:------:|:------:|:-------------------:|:------:|:------:|
> |                  |         | AUC    ↑           | FPR ↓  | HM ↑   | AUC   ↑            | FPR ↓  | HM ↑   | AUC  ↑               | FPR ↓  | HM ↑   |
> | CIFAR‑10         | ZS‑Eval | 96.74             | 18.87  | 73.28  | 92.12             | 43.98  | 71.37  | 91.59               | 24.66  | 72.28  |
> |                  | TPT [4]    | 96.51             | 19.49  | 73.88  | 92.24             | 44.36  | 71.89  | 91.56               | 24.78  | 73.66  |
> |            | AdaND  [1]  | 98\.79     | 5\.97    | 79\.48     | 91\.96     | 39\.30     | 74\.63     | 98\.71        | 4\.30     | 80\.31     |
> |                  | ROSITA [2]  | 99.18             | 3.82   | 81.72  | 93.39             | 33.47  | 74.59  | 98.88               | 4.12   | 80.93  |
> |                  | Open‑IRT | **99.57**         | **1.66** | **82.35** | **93.95**     | **32.19** | **75.78** | **99.11**           | **3.56** | **81.57** |
> | CIFAR‑100        | ZS‑Eval | 89.27             | 60.99  | 46.18  | 78.25             | 79.92  | 44.64  | 72.66               | 61.23  | 45.77  |
> |                  | TPT [4]     | 89.39             | 60.27  | 46.12  | 78.25             | 80.14  | 44.74  | 72.63               | 61.17  | 46.29  |
> |            | AdaND [1]   | 95\.21     | 16\.91    | 57\.64     | 89\.50     | 49\.96     | 48\.55     | 92\.38        | 28\.81     | 55.\.89     |
> |                  | ROSITA [2] | 94.39             | 23.97  | 57.18  | 90.28             | 45.25  | 51.76  | 91.27               | 30.19  | 56.12  |
> |                  | Open‑IRT | **99.38**         | **3.71** | **58.75** | **94.09**     | **27.21** | **52.28** | **93.29**           | **22.30** | **56.97** |
>
>
> ---
>
>
> ### **\[Weakness 2] Main Improvements of the Experiments**
>
> We observe that in some cases (e.g., Table 1 and Table 3), accuracy improvements are limited. This is due to current baseline methods (e.g., ROSITA \[2], OWTTT \[3]) already reaching near-saturated performance on specific datasets (CIFAR-10C, VisDA). However, our experiments consistently show superior performance in most scenarios, as evidenced in the main text and supplementary tables (**Table 7, Table 10, and Table 11**).
>
> Our proposed intermediate domain approach enhances the separation between ID and OOD samples, leading to notable improvements in **FPR95**. For example, in Table 1 (CIFAR-10C to SVHN with CLIP), our method improves **FPR95** by **12.88%** compared to the second-best method. This is attributed to **Hypothesis 2**, where the intermediate domain concept facilitates better ID-OOD distinction.
>
> In **Table 2** and **Table 3**, apart from near-saturated results (with **FPR95** < 10), our method also shows significant **FPR95** improvements. For instance, in the ImageNet-C to MNIST experiment (Table 2), we observe both an accuracy increase and a **7.49** improvement in **FPR95**, further validating the effectiveness of  **Hypothesis 2**.
>
> ---
>
> ### **\[Weakness 3] Structural Complexity and Completeness of the Method**
>
> Thank you for considering the complexity of our approach. While Open-IRT involves multiple components, each is essential and optimized to realize our core innovation: The **score bank** and **GMM** implement **Hypothesis 1**, while the **KNN search** and **loss terms** realize **Hypothesis 2**.
>
> Our design achieves significant performance gains with minimal overhead, including a 7.19\% average reduction in FPR compared to the second-best method (**\[Weakness 1]**) and an inference latency of just 40.15ms (**\[Question 3]**), demonstrating real-time viability. Ablation studies (**Line 311, 317**) further confirm the module's effectiveness and necessity.
>
> This is an efficient trade-off: Modularity enables the novel intermediate domain mechanism, while latency remains suitable for single-image deployment.
>
> ---
>
> ### **\[Question 1] Additional Experimental Comparison and Analysis Regarding AdaND**
>
> Please refer to the table provided in **\[Weak 1]** section for the experimental results.
>
>
> ---
>
> ### **\[Question 2] Hyperparameter Robustness Experiment**
>
> **Hyperparameters B and K:** We validated the robustness of **B** and **K** in **Figure 10** and **Table 8** (Appendix), with an analysis in **Line 779**. We tested values of **B = 64, 128, 256, 512** and **K = 0, 1, 3, 5**, and found that accuracy fluctuations were less than **0.70%**, indicating that reasonable adjustments to these values do not significantly affect results.
>
> **Hyperparameters $\alpha$ and $\lambda$:** The robustness analysis for **$\alpha$** and **$\lambda$** is presented in **Table 9**. As noted in **Line 796**, changes in **$\alpha$ (0.1 to 0.5)** resulted in accuracy fluctuations of less than **0.33%**, and for **$\lambda$ (0.1 to 0.5)**, fluctuations were under **0.57%**.  It is worth noting that some results in **Table 9** even slightly outperform those in the main experiments (**Table 1**, **Table 7**), as we did not perform deliberate hyperparameter tuning. We firmly believe that a robust method should not rely on selecting the optimal hyperparameter configuration but instead be adaptable to a wide range of real-world deployment scenarios.
>
>
> **Thresholds $\mu_I$ and $\mu_O$:** **$\mu_I$** and **$\mu_O$** are not hyperparameters. Defined in **Line 216**, they are dynamic thresholds computed from confidence values, adapting to sample distributions, thus avoiding robustness issues related to fixed hyperparameters.
>
> ---
>
> ### **\[Question 3] Computational Efficiency of the Method**
>
> From an implementation efficiency perspective, the experimental results, including both overall dataset samples and single-sample analysis. Our method demonstrates excellent computational efficiency. With **a single 3090 GPU**, the average execution time is **40.15ms**, making it suitable for lightweight deployment. These results are expected to improve further with advancements in computational power and hardware performance.
>
> ---
>
> | ID        | OOD          | Samples | Total Time (s) | Time per Sample (ms) |
> |----------------|-------------------|---------|----------------|------------------------|
> | CIFAR‑100C     | MNIST             | 20000   | 823.12         | 41.16                  |
> | CIFAR‑100C     | SVHN              | 20000   | 763.93         | 38.20                  |
> | CIFAR‑100C     | Tiny‑ImageNet     | 20000   | 766.14         | 38.31                  |
> | CIFAR‑100C     | CIFAR‑100C        | 20000   | 765.17         | 38.26                  |
> | CIFAR‑10C      | MNIST             | 20000   | 901.36         | 45.07                  |
> | CIFAR‑10C      | SVHN              | 20000   | 779.13         | 38.96                  |
> | CIFAR‑10C      | Tiny‑ImageNet     | 20000   | 767.28         | 38.36                  |
> | CIFAR‑10C      | CIFAR‑100C        | 20000   | 789.26         | 39.46                  |
> | ImageNet‑C     | MNIST             | 20000   | 816.28         | 40.81                  |
> | ImageNet‑C     | SVHN              | 20000   | 807.21         | 40.36                  |
> | VisDA          | MNIST             |100000   |4096.92         | 40.97                  |
> | VisDA          | SVHN              |100000   |4107.22         | 41.07                  |
> | VisDA          | MNISTM            |100000   |4100.85         | 41.01                  |
>
> | Method                 | Mean Inference Time (ms) |
> |------------------------|---------------------------|
> | Open‑IRT               | 40.15                     |
> | ROSITA                 | 41.39                     |
> | TPT                    | 108.52                    |
> | ZS‑Eval (Inference only) | 12.05                  |
> | AdaND                  | 42.88                     |
> ---
>
> ### **Reference**
> [1] Noisy Test-Time Adaptation in Vision-Language Models, ICLR 2025
>
> [2] Sreenivas, Manogna, and Soma Biswas. "Efficient Open Set Single Image Test Time Adaptation of Vision Language Models." arXiv preprint arXiv:2406.00481 (2024).
>
> [3] Li, Yushu, et al. "On the robustness of open-world test-time training: Self-training with dynamic prototype expansion." Proceedings of the IEEE/CVF ICCV. 2023.
>
> [4] Shu, Manli, et al. "Test-time prompt tuning for zero-shot generalization in vision-language models." NeurIPS 35 (2022): 14274-14289.

---

> > ### Comment · Reviewer_c5VY · 2025-08-05
> >
> > The reviewer sincerely thanks the authors for their careful answer to my question.
> >
> > For AdaND, actually their setup is exactly the same as the one in your paper. They also have ID and OOD splits (e.g., ImageNet as ID and SUN as OOD). They just call the open-set setup as Noisy in their paper.
> >
> > Thanks for sharing the results of AdaND. But the setup in your paper is relatively easy to detect OOD in most cases. For example, in Table 2, you use ImageNet-C (ID) and MNIST/SVHN (OOD). It is very easy to detect those numbers in MNIST from the real image in ImageNet. That's why in your results, the AUC is very easy to reach 99% and FPR to 1%. Besides, in this easy case, there is already a clear separation in the embedding space for ID and OOD (Fig 4), and it is easy to construct Middle-Domain. What if in more complex setups, such as the ones used in AdaND (e.g., ImageNet as ID and SUN as OOD)? Will the idea of Middle-Domain still work and be better than AdaND?
> >
> > Finally, I still have concerns about the limited performance improvement, considering the complexity of the proposed method. The variance of the results is also not reported. Considering the result is very close, especially in Table 1, there is only a 0.07 margin, it is also important to know the variance of the proposed method.
> >
> > Based on the above concerns, I would like to maintain the borderline rating.

---

> ### Author Response · Authors · 2025-08-06
> **Generalization to More Challenging OOD Settings**
>
> We sincerely thank the reviewer for the thoughtful comments and for giving us the opportunity to address these concerns. Below we respond point-by-point.
>
> ###  **[1] On Generalization to Harder OOD Settings**:
>
> We agree with the reviewer that using datasets such as MNIST/SVHN makes the OOD detection problem relatively easier due to their large semantic shift from ImageNet-based ID sets. To address this, we conducted additional experiments on a more challenging and realistic setup: **ImageNet-R (ID)** vs **SUN (OOD)**, following the AdaND [1] protocol. We used 10,000 samples for both ID and OOD and evaluated all methods under the same condition. The results are shown below (with five different random seeds):
>
> | Method (ViT‑B16) |      AUROC ↑    |      FPR ↓     |      HM ↑      |
> | :----------: | :------------: | :------------: | :------------: |
> |    ZS‑Eval   |      81.53     |      82.56     |      63.68     |
> |      TPT     |      81.49     |      82.06     |      63.75     |
> |     AdaND    |      83.09     |      79.15     |      66.70     |
> |    ROSITA    |      81.53     |      83.22     |      64.38     |
> | **Open‑IRT** | **83.48±0.19** | **78.57±0.28** | **67.79±0.25** |
>
> As shown, although this setting leads to generally higher FPRs due to increased difficulty, our method **still achieves the best overall performance**, which demonstrates that the proposed Middle-Domain approach generalizes well to more complex ID-OOD distributions.
>
> We further emphasize that our method performs **better in complex scenarios**, such as CIFAR-100 → Tiny-ImageNet (**Rebuttal Table above**), because the semantic distribution shift allows the Middle-Domain to function more effectively. Even when the semantic space is shared and only the style differs (i.e., in Domain Adaptation settings), Middle-Domain representations can still be constructed effectively, as shown in prior work \[2]. Thus, **as long as there exists any semantic or distributional gap between ID and OOD, our method is applicable**, and its advantage becomes even more prominent with increasing difficulty (see **Line 295**).
>
> ---
>
> ###  **[2] On Performance Margin**:
>
> We understand the reviewer’s concern that the performance gain in Table 1 may appear modest. This is due to the relative simplicity of datasets such as CIFAR-10C, where many methods already achieve strong results. In contrast, on harder datasets (**Table 2** and **Table 3**, e.g., VisDA or ImageNet-C), the improvements become much more substantial:
>
> > For example, in the ImageNet-C → SVHN experiment, Open-IRT achieves a **+5.12%** increase in Acc\_HM, **-16.86%** in FPR95, and **+6.55%** in AUROC.
>
> This demonstrates that **our approach is particularly beneficial when the feature space is complex or less separable**, which is where many real-world TTA scenarios lie.
>
> Moreover, even on easier datasets, we still show consistent performance:
>
> > In CIFAR-10C (Table 1), across 8 groups, Open-IRT achieves an average AUC improvement of **+0.20%**, FPR reduction of **-1.92%**, and HM accuracy gain of **+0.44%** over the second-best method — without extensive hyperparameter tuning.
>
> Based on our existing experimental results, we further report the variance across five random seeds in **Table 1** to demonstrate the stability of our method.
>
> | HM Accuracy (CLIP-based Open‑IRT) |      MNIST     |      SVHN      |  Tiny-ImageNet | CIFAR-100C / CIFAR-10C |
> | :-------------------------------: | :------------: | :------------: | :------------: | :--------------------: |
> |         CIFAR-10C                           | 84.53±0.09 | 80.57±0.12 | 81.03±0.11 |     69.20±0.08    |
> |         CIFAR-100C                         | 62.28±0.27 | 56.28±0.21 | 56.60±0.26 |     46.94±0.29     |
>
> The low variance and consistent improvements across settings further reinforce the robustness of our method.
>
> ---
>
> ###  **[3] On Computational Complexity:**
>
> As reported in the rebuttal, our method is efficient in practice: it takes **only 40ms per image on a single 3090 GPU**, and scales better with higher-end hardware. Open-IRT is designed as a lightweight feature-level enhancement, and we believe its benefits outweigh its minor overhead, especially in real-world TTA applications.
>
> ---
>
> ###  **[4] On Complementarity to Existing Methods**:
>
> We emphasize that the Middle-Domain theory is **orthogonal and complementary** to existing OOD-TTA frameworks. It can be integrated into other methods to potentially boost their performance further.  **We sincerely thank the reviewer again for the insightful suggestions. We will include all additional experiments, variance results, and discussions in the revised paper and supplementary.**
>
> ---
>
> ### **References**
> [1] Noisy Test-Time Adaptation in Vision-Language Models, ICLR 2025
>
> [2] Huang et al. "Arbitrary style transfer in real-time with adaptive instance normalization." ICCV 2017
>
> ---

---

> > ### Comment · Reviewer_c5VY · 2025-08-07
> >
> > Thanks to the authors for their new results and analysis. Most of my concerns have been addressed. I will make the final rating based on these updates.

---

> > > ### Author Response · Authors · 2025-08-07
> > >
> > > Thank you for your comments. We are glad that our additional results and analysis have addressed most of your concerns. We sincerely appreciate your time and effort in reviewing our work. **Wishing you all the best in your research and daily life.**

---

### Author Response · Authors · 2025-08-05
**Follow-up and Clarification on Rebuttal for Paper**

Dear Reviewers,

We sincerely thank all reviewers for your recognition of the novelty of our proposed method, as well as the positive comments from reviewers c5VY, 4p14, and g3am on the clarity of our writing and overall presentation.

In response to your suggestions and questions, we have provided additional clarifications and supporting results during the discussion phase, including further comparisons, and an efficiency study to more comprehensively address your concerns.

As the discussion deadline approaches, we would like to respectfully ask whether there are any remaining concerns or questions regarding our response. We would be happy to provide further clarification if needed.

Thank you again for your time and constructive insights. Wishing you all the best in your research and daily life.

Warm regards,

All authors

---

### Note · Authors · 2025-08-11

Dear Reviewers, ACs, and Conference Organizers,

**We sincerely thank you all for your time, effort, and constructive feedback. We truly appreciate the reviewers' recognition of the novelty and practical value of our method.**

During the discussion stage, we made every effort to address all raised concerns and are pleased that **the reviewers carefully and responsibly** considered our responses, resulting in positive outcomes. We deeply value the reviewers’ expertise and thoughtful reading of our work, and remain committed to incorporating their suggestions in the revised version to further enhance the clarity and rigor of the paper.

Once again, we extend our sincere appreciation to everyone involved for their dedication and hard work throughout this process.

Warm regards,

All authors

---

### Decision · Program_Chairs · 2025-09-17

**Decision:**

Accept (poster)

**Comment:**

Authors introduce in this work a challenging and practical scenario for Test-Time adaptation of pre-trained vision language models, which deals with an open-set single-image problem. The paper initially received mixed scores, with reviewers highlighting the problem as under explored, and acknowledging the novelty and soundness of the proposed approach, whose practical benefits (efficiency and versatility) are supported by comprehensive experiment results. Reviewers also raised important concerns, including unclear technical presentation, insufficient explanation of key design choices, lack of comparison to relevant test-time adaptation methods, or the inclusion of more complex datasets, among others. After carefully reading the discussions, authors have successfully addressed the majority of these concerns, and reviewers are satisfied with the authors responses.

Considering all these points, I recommend the acceptance of this work. Having said this, I strongly encourage the authors to consider one remaining concern from one reviewer, i.e., revisiting the claim of single-image operation. The proposed approach relies on feature and score banks that accumulate information over multiple samples, and the single-image operation claim might sound a bit overstated. Similarly, despite these claims, the supplementary ablation study on the batch size does not explore values B<64, which does not support the single-image claim.